# Speed variations of bacterial replisomes

**Deepak Bhat[1,2], Samuel Hauf[3], Charles Plessy[4], Yohei Yokobayashi[3], Simone Pigolotti[1]\***

[1]Biological Complexity Unit, Okinawa Institute of Science and Technology, Onna, Japan; [2]Department of Physics, School of Advanced Sciences, Vellore Institute of Technology, Vellore, Tamil Nadu, India; [3]Nucleic Acid Chemistry and Engineering Unit, Okinawa Institute of Science and Technology, Onna, Japan; [4]Genomics and Regulatory Systems Unit, Okinawa Institute of Science and Technology, Onna, Japan

**Abstract** Replisomes are multi-protein complexes that replicate genomes with remarkable speed and accuracy. Despite their importance, their dynamics is poorly characterized, especially in vivo. In this paper, we present an approach to infer the replisome dynamics from the DNA abundance distribution measured in a growing bacterial population. Our method is sensitive enough to detect subtle variations of the replisome speed along the genome. As an application, we experimentally measured the DNA abundance distribution in *Escherichia coli* populations growing at different temperatures using deep sequencing. We find that the average replisome speed increases nearly fivefold between 17 °C and 37 °C. Further, we observe wave-like variations of the replisome speed along the genome. These variations correlate with previously observed variations of the mutation rate, suggesting a common dynamical origin. Our approach has the potential to elucidate replication dynamics in *E. coli* mutants and in other bacterial species.

## Editor's evaluation

This manuscript combines theory with experiments to characterize the replication speed of bacteria chromosomes through the cell cycle. The authors show oscillatory patterns in the replication speed of *E. coli*, which they relate to the heterogeneity of mutation rates along the genome, suggesting a tradeoff between the speed and accuracy of replication. This work presents an elegant approach for investigating bacterial growth from a systems biology perspective.

**\*For correspondence:**
simone.pigolotti@oist.jp

## Introduction

Every cell must copy its genome in order to reproduce. This task is carried out by large protein complexes called replisomes. Each replisome separates the two DNA strands and synthesizes a complementary copy of each of them, thereby forming a Y–shaped DNA junction called a replication fork. The speed and accuracy of replisomes is impressive (*Baker and Bell, 1998*). They proceed at several hundreds to one thousand base pairs per second (*Pham et al., 2013*; *Elshenawy et al., 2015*), with an inaccuracy of about one mis-incorporated monomer every 10 billion base pairs (*Schaaper, 1993*). In bacteria, two replisomes initiate replication at a well-defined origin site on the circular genome, progress in opposite directions, and complete replication upon encountering each other in a terminal region.

The initiation and the completion of DNA replication conventionally delimit the three stages of the bacterial cell cycle (*Dewachter et al., 2018*; *Wang and Levin, 2009*). The first stage, B, spans the period from cell birth until the initiation of DNA replication. The second stage, C, encompasses the time needed for replication. The last phase, D, begins at the end of DNA replication and concludes with cell division. While it is established that DNA replication and the cell cycle must be coordinated,

their precise relation has been a puzzle for decades (*Willis and Huang, 2017*). A classic study by *Cooper and Helmstetter, 1968* finds that, upon modifying the growth rate by changing the nutrient composition in *Escherichia coli*, the durations of stages C and D remain constant at about 40 min and 20 min, respectively. This means that the replisome speed must be unaffected by the nutrient composition, at least on average. When the cell division time is shorter than one hour, DNA replication is initiated in a previous generation. This implies that, in fast growth conditions, multiple pairs of replisomes simultaneously replicate the same genome (*Fossum et al., 2007*). Tuning the growth rate by changing the temperature has a radically different effect on bacterial physiology. For example, in vivo (*Pierucci, 1972*) and in vitro (*Yao et al., 2009*) studies show that the speed of replisomes is affected in this case.

More precise features of replisome dynamics, such as whether their speed is approximately constant or varies along the genome, are important to determine the location of their encounter point and the distribution of replication errors on the genome (*Niccum et al., 2019*; *Dillon et al., 2018*). However, this detailed information is hard to obtain (*Pham et al., 2013*). One way for inferring it is to measure the DNA abundance distribution, that is the frequency of DNA fragments along the genome in an exponentially growing cell population. In fact, the frequency of these fragments in the population depends on the proportions of synthesizing genomes of different lengths, which in turn depend on the replisome dynamics. Previous studies have used the DNA abundance distribution to understand the functioning of bacterial replication and how different proteins assist completion of DNA replication (*Wendel et al., 2014*; *Wendel et al., 2018*; *Rudolph et al., 2013*; *Midgley-Smith et al., 2019*; *Midgley-Smith et al., 2018*). However, these studies focused on qualitative analysis of the observed changes of the DNA distribution in knockout mutants with respect to the wild type, and did not attempt to predict the shape of the distribution using quantitative theoretical models. The DNA abundance distribution has also been used to identify actively growing species in a microbiome (*Korem et al., 2015*).

In this paper, we introduce a method to infer the replisome dynamics from the DNA abundance distribution. As an application, we experimentally measured the DNA abundance distribution of *E. coli* growing at different temperatures between $17^oC$ and $37^oC$ using high-throughput sequencing. Our approach, combined with our experiments, shows that the average speed of replisomes exhibits an Arrhenius dependence on the temperature, with an almost fivefold variation in the range we considered. Moreover, the precision of our experiments reveals that the speed of replisomes varies along the genome in a seemingly periodic and highly repeatable fashion around this average value. We find that this pattern is highly correlated with previously observed wave-like variations of the single base pair mutation rate along the bacterial genome (*Niccum et al., 2019*; *Dillon et al., 2018*). We discuss possible common causes for these two patterns.

## Results

### Distribution of genome types

We consider a population of bacteria that grow exponentially in a steady environment. Each cell in the growing population can encompass three types of genomes, see *Figure 1a* and *Figure 1b*: (i) one template genome, that is, the genome that the cell inherited at its birth. (ii) incomplete genomes, that is, genomes which are being synthesized. (iii) post-replication genomes that will be passed to new cells and become their templates.

In nutrient-rich conditions, bacteria replicate their genome in parallel, so that the numbers of incomplete genomes and post-replication genomes per cell are variable, see *Figure 1b*. The classic Cooper-Helmstetter model (*Cooper and Helmstetter, 1968*) describes the dynamics of these genomes in a given cell through generations. We adopt a different approach and focus on the abundance dynamics of the three types of genomes in the whole population. We call $N_T(t)$, $N_S(t)$, $N_P(t)$ the total number of template genomes, incomplete (synthesizing) genomes, and post-replication genomes, respectively, that are present in the population at time $t$. Our first aim is to quantify the relative fractions of these three types of genomes.

The total number of genomes is $N(T) = N_T(t) + N_S(t) + N_P(t)$. Since each cell contains exactly one template, the total number of cells is equal to $N_T(t)$. The total number of genomes evolves as effect of: (a) replication initiation, which creates new synthesizing genomes at a rate $k$; (b) completion of replication, which transforms synthesizing genomes into post-replication ones at rate $\beta$; and (c) cell division,

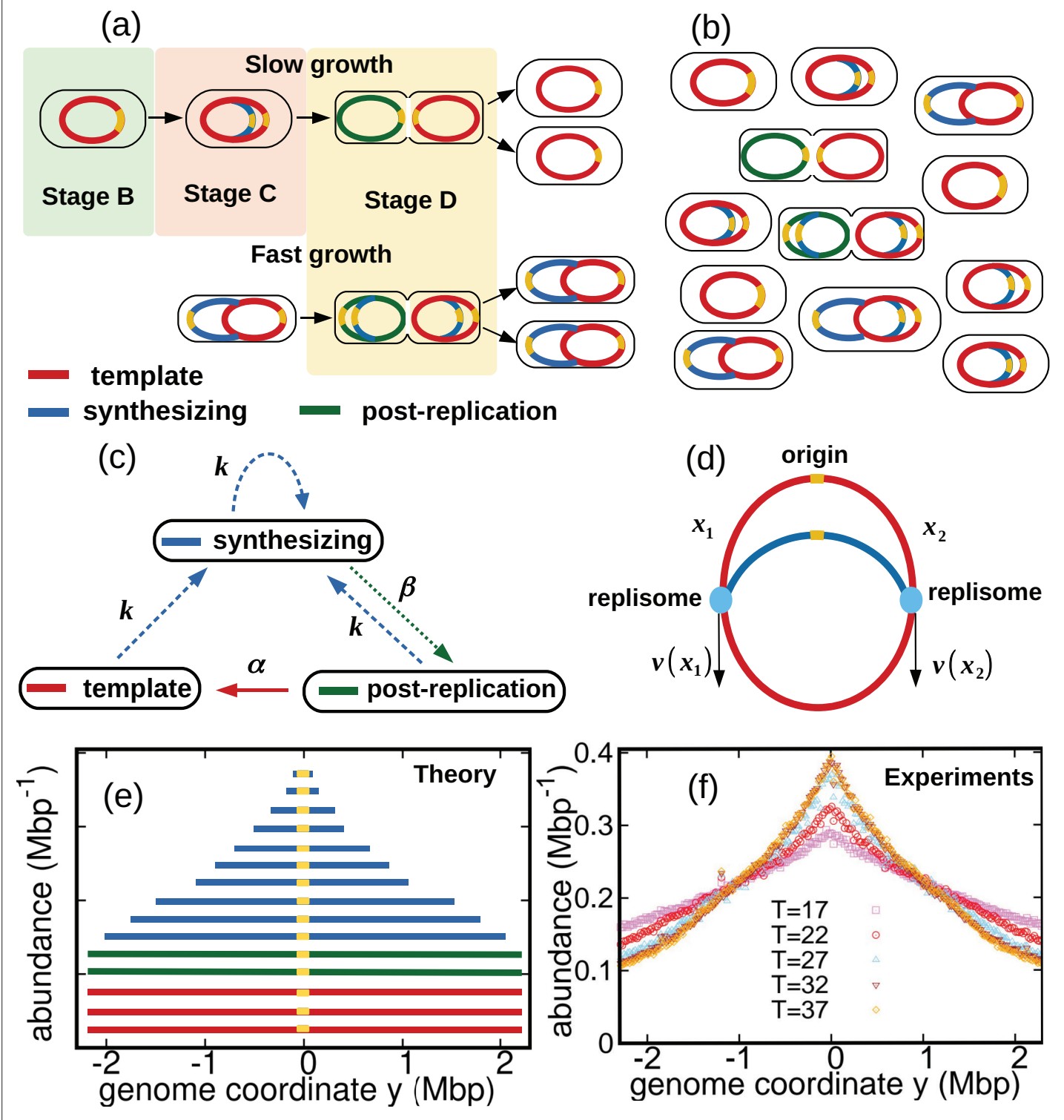

**Figure 1.** Dynamics of genome types and DNA abundance distribution in an exponentially growing bacterial population. (**a**) Cell cycle. In slow growth conditions (top panel), newborn cells contain a template (stage B, red). As the cell cycle progresses, two replisomes synthesize a new genome (stage C, blue) starting from the origin on the template (yellow spot). When replication terminates, cells contain the original template and a post-replication genome (stage D, green). Upon subsequent cell division, the post replication genome becomes the template for the newborn cell. In fast growth conditions (bottom panel), newborn cells acquire a template which is already undergoing synthesis. In subsequent stages, multiple replicating genomes may exist in the same cell. (**b**) Composition of genomes in an exponentially growing population of cells. Each cell may contain a different number of genomes, depending on its stage in the cell cycle and growth conditions. (**c**) Dynamics of genome types. Dashed blue arrow represent initiation of

*Figure 1 continued on next page*

*Figure 1 continued*

replication. The dotted green arrow represents completion of replication. The solid red arrow represents cell division. (**d**) Replisome dynamics. Two replisomes begin replication at an origin and progress in opposite directions. Their speed may depend, in general, on their genome coordinate. (**e**) Sketch of the DNA abundance distribution as a function of the genome coordinate. All three types of genomes contribute to the DNA abundance distribution. Because of incomplete genomes, the DNA abundance is largest at the origin and smallest at the terminal region (i.e., towards the periphery). (**f**) Experimental DNA abundance distribution at different temperatures.

which turns post-replication genomes into templates at a rate α, see *Figure 1c*. This dynamics is described by the set of equations:

$$\frac{d}{dt}N_T(t) = \alpha N_P \tag{1}$$

$$\frac{d}{dt}N_S(t) = kN - \beta N_S \tag{2}$$

$$\frac{d}{dt}N_P(t) = \beta N_S - \alpha N_P. \tag{3}$$

It follows from *Equations 1–3* that, in steady growth, the total number of genomes grows exponentially at a rate equal to the fork firing rate $k$. In this exponential regime, the fractions of the three genome types are constant:

$$\frac{N_T(t)}{N(t)} = \frac{\alpha\beta}{(k+\beta)(k+\alpha)} \tag{4}$$

$$\frac{N_S(t)}{N(t)} = \frac{k}{(k+\beta)} \tag{5}$$

$$\frac{N_P(t)}{N(t)} = \frac{\beta k}{(k+\beta)(k+\alpha)}. \tag{6}$$

The ratio $N/N_T$ can be interpreted as the average number of genomes per cell. Since this ratio is constant, the fork firing rate $k$ can also be identified as the exponential growth rate of the number of cells. For this reason, from now on, we refer to $k$ as the 'fork firing rate' or the 'growth rate' interchangeably.

In principle, the rates α, β, and $k$ should depend on the 'age' of each genome, that is the time spent by the genome in each stage. In Appendix 1, we generalize our model to an age-dependent model to account for this fact. We find that this age-dependent model is equivalent to *Equations 1–3* in the exponential growth regime. This result supports the use of our simple model of genome-type dynamics.

We now analyze the incomplete genomes in more detail. We call $x_1$ and $x_2$ the portions of a given incomplete genome copied by the two replisomes at a given time, with $0 \leq x_1, x_2 \leq L$, see *Figure 1d*. Replication initiates at $x_1 = x_2 = 0$ and completes once the replisomes meet each other, that is, $x_1 + x_2 = L$. The replisome dynamics proceeds as follows. Each replisome is characterized by a speed which depends, in principle, on the replisome position (be it $x_1$ or $x_2$) and by a diffusion coefficient representing random fluctuations of the speed. The coordinates $x_1, x_2$ of the two replisomes evolve according to the stochastic differential equations:

$$\begin{aligned} \frac{d}{dt}x_1(t) &= v(x_1) + \sqrt{2D}\,\xi_1(t) \\ \frac{d}{dt}x_2(t) &= v(x_2) + \sqrt{2D}\,\xi_2(t), \end{aligned} \tag{7}$$

where $\xi_1(t)$ and $\xi_2(t)$ are white noise variables satisfying $\langle\xi_1(t)\rangle = \langle\xi_2(t)\rangle = 0$, $\langle\xi_1(t)\xi_1(t')\rangle = \langle\xi_2(t)\xi_2(t')\rangle = \delta(t - t')$, and $\langle\xi_1(t)\xi_2(t')\rangle = 0$. Here, $\langle\ldots\rangle$ denotes an average over realizations.

Close to thermodynamic equilibrium, the diffusion coefficient $D$ can be estimated by the Stokes-Einstein relation (*Hynes, 1977*). However, since replisomes are driven far from equilibrium by hydrolysis of dNTPs, their diffusion coefficient could deviate from this estimate. In the absence of fluctuations ($D = 0$), each of the two replisomes copies exactly half of the genome, whereas for $D > 0$ their meeting point is characterized by a certain degree of uncertainty.

In steady exponential growth, we call $p^{\text{st}}(x_1, x_2)$ the stationary probability distribution of finding an incomplete genome with copied portions $x_1$ and $x_2$. This probability distribution satisfies the equation:

$$\vec{\nabla} \cdot [\vec{v} p^{\text{st}}] + D\nabla^2 p^{\text{st}} - kp^{\text{st}} = 0 , \tag{8}$$

where we introduce the vector notation $\vec{v} = (v(x_1), v(x_2))$ and $\vec{\nabla} = (\partial/\partial x_1, \partial/\partial x_2)$. The last term in the right hand side of *Equation 8* is a dilution term that accounts for the exponential increase in newborn cells. We formally derive *Equation 8* and discuss its boundary conditions in Methods.

## DNA abundance distribution

The DNA abundance distribution $\mathcal{A}(y)$ is the probability that a small DNA fragment randomly picked from the population originates from genome position $y$, see *Figure 1e*. We define the genome coordinate $y \in [-L/2, L/2]$, where $y = 0$ corresponds to the origin of replication and $L$ is the genome length. Templates and post-replication genomes yield a uniform contribution to the distribution $\mathcal{A}(y)$ (red and green in *Figure 1e*). In contrast, incomplete genomes contribute in a way that depends on the replisome positions along the genomes (blue in *Figure 1e*). Our experiments permit to measure the distribution $\mathcal{A}(y)$ with high accuracy, see *Figure 1f* and Methods.

To express the distribution $\mathcal{A}(y)$ in our model, we first introduce the probability $\mathcal{P}(y)$ that a randomly chosen genome (either complete or incomplete) in the population includes the genome location $y$. This probability is expressed by

$$\mathcal{P}(y) = \frac{k}{\beta + k} \overbrace{\left[ \int_{|y|}^{L} dx_1 \int_{0}^{L-x_1} dx_2 \, p^{\text{st}}(x_1, x_2) + \int_{L-|y|}^{L} dx_2 \int_{0}^{L-x_2} dx_1 \, p^{\text{st}}(x_1, x_2) \right]}^{y \text{ in an incomplete genome}} + \overbrace{\frac{\beta}{k + \beta}}^{y \text{ in a complete genome}} , \tag{9}$$

where we assumed that the dynamics of the two replisomes is symmetric, so that $p^{\text{st}}(x_1, x_2) = p^{\text{st}}(x_2, x_1)$, and we used that a randomly chosen genome is complete with probability $(1 - N_S/N) = \beta/(k + \beta)$, see *Equation 5*. The DNA abundance distribution $\mathcal{A}(y)$ is proportional to $\mathcal{P}(y)$, up to a normalization constant:

$$\mathcal{A}(y) = \frac{\mathcal{P}(y)}{\int_{-L/2}^{L/2} \mathcal{P}(y')dy'} . \tag{10}$$

For given choices of $v(x)$, $D$, and $k$, our theory permits to compute the distribution of incomplete genomes $p^{\text{st}}(x_1, x_2)$ via *Equation 8*. From this solution, we can also calculate $\beta$ as the steady rate at which replisomes complete replication (see Methods). This information can be used to compute the DNA abundance distribution $\mathcal{A}(y)$ using *Equation 10*. Therefore, by experimentally measuring the DNA abundance distribution, we can test our hypotheses on the speed function $v(x)$ and the diffusion coefficient $D$.

## Constant speed model

We first consider a scenario in which replisomes progress at a constant speed $\bar{v}$ and without fluctuations, $D = 0$. We find that, in this case, the DNA abundance distribution is expressed by

$$\mathcal{A}(y) = \frac{k}{2\bar{v}[1 - e^{-kL/2\bar{v}}]} e^{-k|y|/\bar{v}} , \tag{11}$$

see Methods. We fit this distribution to the experimental data using maximum likelihood, see *Figure 2a*. The speed $\bar{v}$ is the only fitting parameter, because we independently measure the exponential growth rate $k$ from the optical density, see Methods.

We find that the speed increases nearly fivefold with temperature in the range we considered and appears to follow an Arrhenius law, see *Figure 2b*. This behavior resembles that of the growth rate. The effective activation energy characterizing the cell cycle is larger than that characterizing the replisome speed, see *Figure 2c*, possibly due to the large number of molecular reactions involved in the cell cycle. The data point at 17°C appears to deviate from the Arrhenius law for both the speed and the growth rate (*Roy et al., 2021*), see *Figure 2c*.

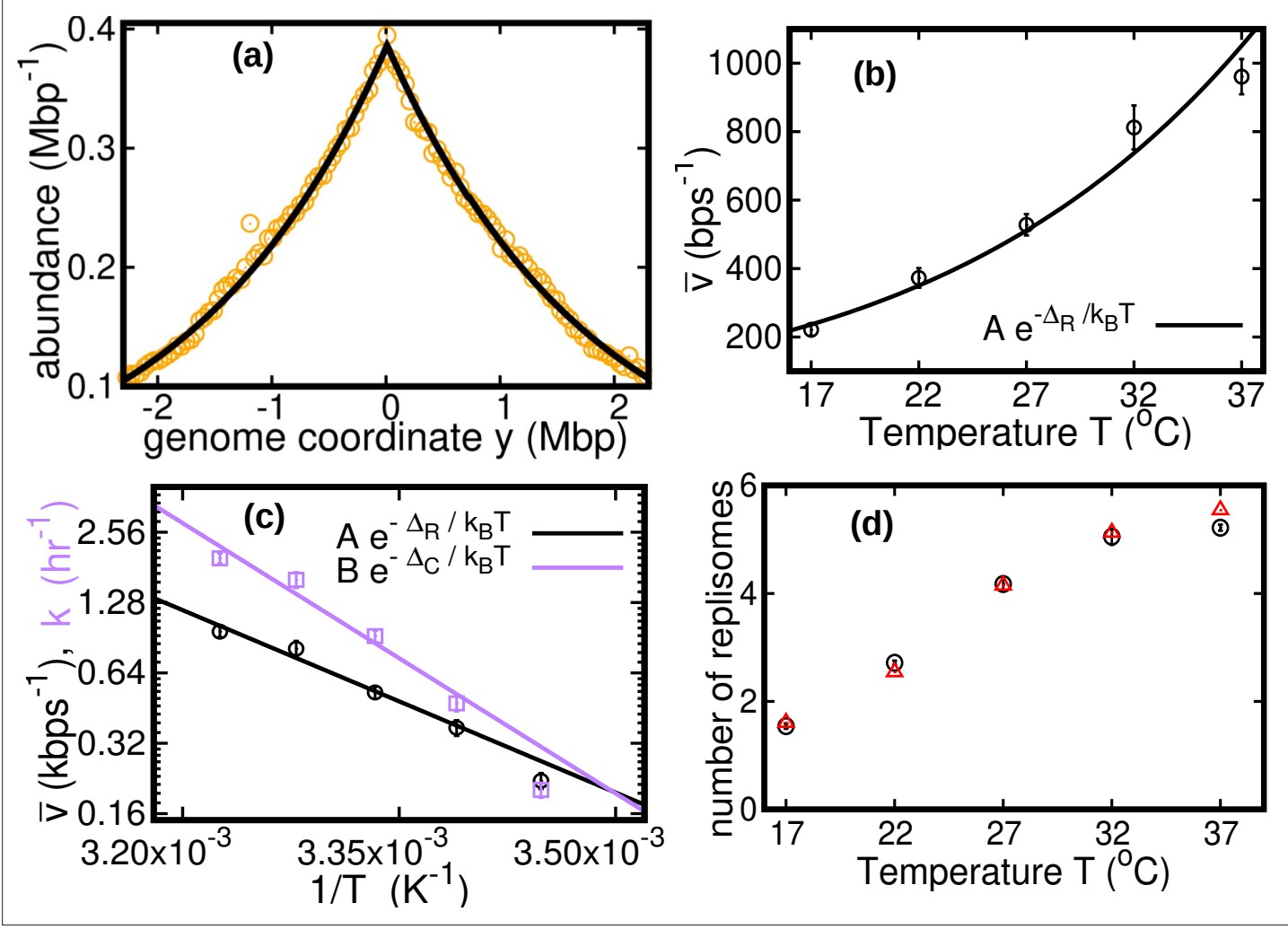

**Figure 2.** Results of the constant speed model. (**a**) DNA abundance distribution for $T = 37^{\circ}C$. Orange circles represent experimental data. The solid black line is the prediction of our model assuming constant speed and $D = 0$. Fits are performed using a maximum likelihood method, see Appendix 2 for details. The quality of fits for replicates and other temperatures is comparable, see *Figure 2—figure supplement 1*, *Figure 2— figure supplement 2*, and *Figure 2—figure supplement 3*. In particular, fits of replicates yield similar values of the speed $\bar{v}$. (**b**) Replisome speed as a function of temperature. Error bars represent sample-to-sample variations. (**c**) Comparison of the temperature-dependence of speed and growth rate (see Methods for details on the growth rate estimation). The solid curves are fits of Arrhenius laws to the data. The fitted parameters are $A = (2.5 \pm 5.3) \times 10^{8}$ bp s$^{-1}$, $\Delta_R = (50 \pm 5)$ kJmol$^{-1}$, $B = (6.0 \pm 24.9) \times 10^{12}$ hr$^{-1}$ and $\Delta_C = (74 \pm 10)$ kJmol$^{-1}$. We exclude the data point for $T = 17^{\circ}C$ in both fits. (**d**) Estimated number of replisomes per complete genome at different temperatures. The red triangles represents the estimate from *Equation 19* in which we use the expression of $\beta$ for the constant speed model, *Equation 22*. The black circles are the estimates from *Equation 20*.

The online version of this article includes the following figure supplement(s) for figure 2:

**Figure supplement 1.** Fits for replicates and other temperatures (normalization 1).

**Figure supplement 2.** Fits for replicates and other temperatures (normalization 2).

**Figure supplement 3.** Fits for replicates and other temperatures (normalization 3).

As seen in *Figure 2c*, the replisome speed does not increase as fast as the growth rate at increasing temperature. This observation suggests that the number of replisomes per genome should increase with temperature to compensate for this gap. Indeed, in the temperature range we studied, our model predicts that the average number of replisomes per complete genome increases from two to almost six, see *Figure 2d* and Methods.

We now focus on the average genome content per cell. Since the model assumes that genomes evolve independently, the average DNA content per cell $\mathcal{C}$ is the product of the average genome

length $\ell$ times the average number of genomes per cell $N/N_T$. Computing the average genome length in the model (see Methods) and using **Equation 4** for the average number of genomes per cell, we obtain that the average DNA content per cell is expressed by

$$C = \frac{2\bar{v}}{k} \frac{k + \alpha}{\alpha} \left[ e^{kL/(2\bar{v})} - 1 \right]. \tag{12}$$

The classic Cooper-Helmstetter model (**Cooper and Helmstetter, 1968**) predicts the DNA content per cell assuming constant durations of stages B, C, and D of the cell cycle. Since we assumed constant speed and $D = 0$, the duration of the replication cycle $L/(2\bar{v})$ is constant in our case as well. As a consequence, the prediction of **Equation 12** is equivalent to that of the Cooper-Helmstetter model (see Appendix 3).

It is generally believed that the ratio between the average DNA content per cell $C$ and the protein content per cell should be maintained approximately constant at varying physiological conditions. This implies that $C$ should be proportional to the cell size. If the growth rate is varied by changing the nutrient composition, $\bar{v}$ remains constant (**Cooper and Helmstetter, 1968**). **Equation 12** then predicts an approximately exponential growth of $C$ with $k$, which is consistent with observations. In this case, the Schaechter–Maaloe–Kjeldgaard growth law states that the cell size grows exponentially with $k$ (**Schaechter et al., 1958**), thereby ensuring DNA-protein homeostasis. In the case of varying temperature, we find that $\bar{v}$ and $k$ present a similar dependence on $T$ (see **Figure 2c**), so that their ratio and thereby $C$ weakly depends on $k$ (see **Appendix 3—figure 1**). Our result is consistent with observations showing that, at increasing temperature, the cell size remains approximately constant (**Shehata and Marr, 1975**) or possibly slightly increases (**Trueba et al., 1982**).

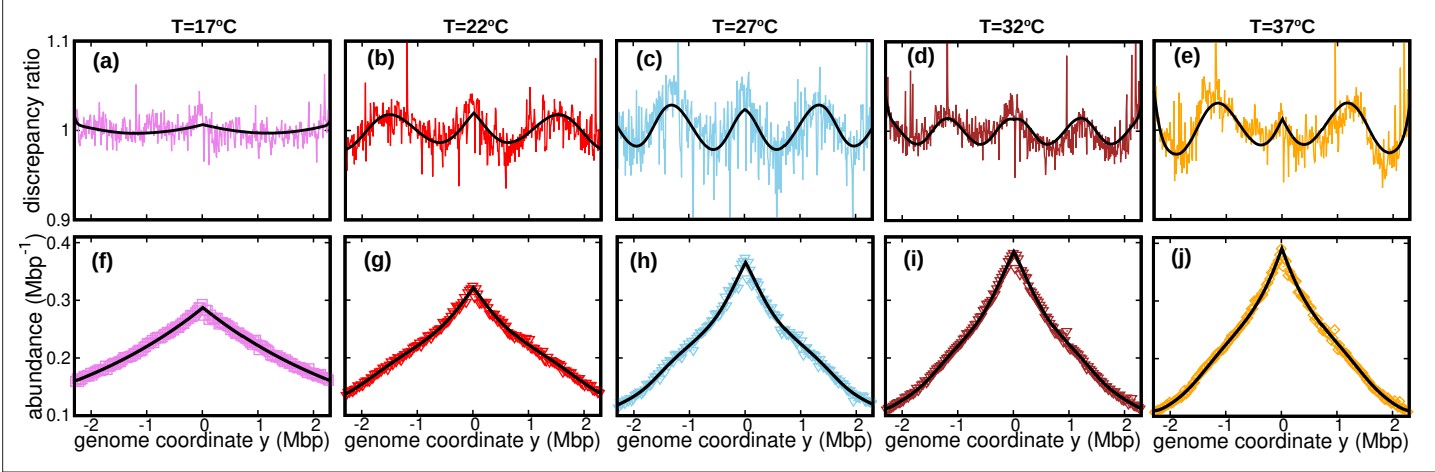

**Figure 3.** Wave-like deviations from the predictions of the constant speed model. (**a–e**) Colored lines: ratios of the experimental DNA abundance over the corresponding prediction assuming constant speed and $D = 0$. The solid black lines represent the ratios of the predictions assuming oscillatory speed, **Equation 13** and $D \geq 0$, over constant speed and $D = 0$. Corresponding plots for replicates and other temperatures are presented in **Figure 3—figure supplement 1**, **Figure 3—figure supplement 2** and **Figure 3—figure supplement 3**. (**f–j**) Experimental DNA abundance distribution at different temperatures. The solid black lines are the fits of the oscillatory speed model. Tests based on the Akaike information criterion show that the oscillatory speed model should be chosen over the constant speed model for all the replicates and at all temperatures, see **Figure 3— figure supplement 5**. The fitted parameters are reported in **Table 1**.

The online version of this article includes the following figure supplement(s) for figure 3:

**Figure supplement 1.** Discrepancy ratio for replicates (normalization 1).

**Figure supplement 2.** Discrepancy ratio for replicates (normalization 2).

**Figure supplement 3.** Discrepancy ratio for replicates (normalization 3).

**Figure supplement 4.** Discrepancy ratio computed from previous data (**Midgley-Smith et al., 2018**).

**Figure supplement 5.** Comparison between the constant speed model and oscillatory model.

**Table 1.** Parameters of the oscillatory speed model.
Temperatures are expressed in $^{\circ}$C, $\bar{v}$ in $\mathrm{bp\,s^{-1}}$, $\omega$ in $\mathrm{rad\,Mbp^{-1}}$, $\phi$ in rad, and $D$ in $\mathrm{kbp^2 s^{-1}}$. Reported values are averages and standard deviations over experimental replicates. The oscillatory and constant speed models yield estimates of the parameter $\bar{v}$ that are consistent with each other, see *Appendix 4—table 1*.

| T | $\bar{v}$ | δ | ω | $\phi$ | D |
|---|---|---|---|---|---|
| 17 | 246±33 | 0.22±0.13 | 0.7±0.5 | 3.3±1.0 | 0.39±0.43 |
| 22 | 351±30 | 0.20±0.06 | 2.7±0.6 | 3.4±0.6 | 0.81±1.18 |
| 27 | 541±30 | 0.18±0.03 | 4.7±0.1 | 2.1±0.1 | 0.35±0.49 |
| 32 | 821±66 | 0.11±0.04 | 5.5±0.2 | 1.5±0.1 | 1.15±1.23 |
| 37 | 970±51 | 0.17±0.03 | 4.3±0.2 | 3.0±0.2 | 2.90±2.48 |

## Oscillating speed model

The assumption of constant speed leads to a rather good fit of our DNA abundance data. However, the precision of our data permits us to appreciate systematic deviations from the model predictions under the constant speed hypothesis, see *Figure 3a-e*. These deviations appear as regular oscillations as a function of the genome coordinate. They are evident at all the temperatures we studied except for 17$^{\circ}$C, where they are barely visible. They are highly repeatable (see *Figure 3—figure supplement 1*, *Figure 3—figure supplement 2* and *Figure 3—figure supplement 3*) and approximately symmetric with respect to the origin of replication. We also analyzed previous experimental data from *Midgley-Smith et al., 2018*. We observed clear oscillations for experiments in nutrient-rich LB medium, but not for experiments in M9 minimal medium, see *Figure 3—figure supplement 4*. This analysis further supports that this phenomenon is robust, at least in fast growth conditions.

To account for these observations, we introduce a more refined model in which the replisome speed oscillates along the genome:

$$v(x) = \bar{v}[1 + \delta \cos(\omega x + \phi)], \tag{13}$$

where δ represents the relative amplitude of oscillations; $\omega$ their angular frequency along the genome; and $\phi$ their initial phase. We also take into account random speed fluctuations in this case, $D \geq 0$. In this case, we predict the DNA abundance distribution using stochastic simulations, see Methods. By fitting the DNA abundance, we estimate the parameters $\bar{v}$, δ, $\omega$, $\phi$, and $D$, see *Figure 3(f–j)* and *Table 1*.

Our fitted speed oscillations are reminiscent of a previously observed wave-like pattern in the mutation rate along the genome of different bacterial species (*Dillon et al., 2018*; *Niccum et al., 2019*). For a quantitative comparison, we analyze this pattern in a mutant *E. coli* strain lacking DNA mismatch repair (*Niccum et al., 2019*). We find that the oscillations in mutation rate and speed are highly correlated, see *Figure 4a*. The mutation rate appears approximately in phase with the speed, meaning that regions where replisomes proceed at higher speed are characterized by a higher mutation rate. This observation leads to the hypothesis that the two phenomena have a common cause.

We consider two possible explanations for these oscillations. The first is that the oscillations originate from a systematic process related with the cell cycle (*Niccum et al., 2019*). The second explanation is that the oscillations are caused by competition among replisomes for nucleotides or other molecules required for replication. Assuming approximately constant cell division times, we estimate the cell division time as $\tau = (\ln 2)/k$. Since $k$ is also equal to the fork firing rate per genome, the time between firing events in a cell is also approximately equal to $\tau$, so that the two hypotheses lead to the same quantitative prediction. If the speed of replisomes was coupled to a factor oscillating with period $\tau$, this would cause spatial oscillation of speed with angular frequency $\omega = 2\pi/(\bar{v}\tau) = 2\pi k/[(\ln 2)\bar{v}]$. This prediction qualitatively agrees with our fitted values of $\omega$, see *Figure 4b*.

If the wave-like pattern were caused by competition among replisomes, one would expect either a minimum of the speed every time a new fork is fired ($\phi = \pi$) or the speed to start decreasing when a new fork is fired ($\phi = \pi/2$). Our fitted values of the phase $\phi$ are also compatible with this range, see *Figure 4c*.

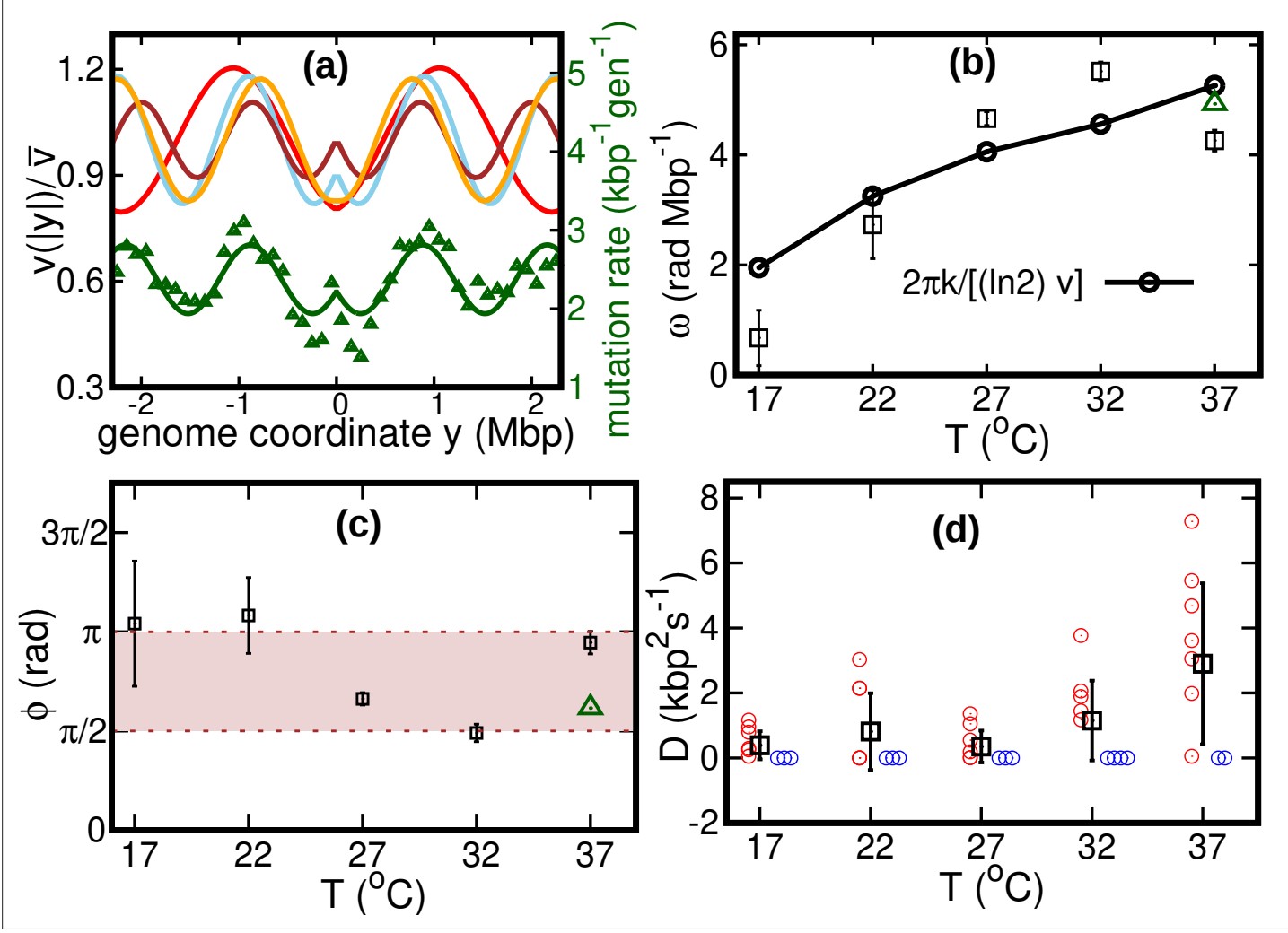

**Figure 4.** Results of the oscillatory speed model. (**a**) Solid lines: relative speeds $v(|y|)/\bar{v}$ along the genome (Red: T = 22°C, sky blue: T = 27°C, brown: T = 32°C, and orange: T = 37°C). We omitted the curve for T = 17°C as the oscillations are less evident in this case (see *Figure 4—figure supplement 1*). The wave-like pattern of the speed is quantitatively similar to the variations of the mutation rate along the genome (green triangles, from *Niccum et al., 2019*; Pearson correlation coefficients between speed and mutation rate: $r_{22C} = 0.42$; $r_{27C} = 0.84$; $r_{32C} = 0.80$ and $r_{37C} = 0.69$). The mutation rate is defined as the number of base pair substitutions per generation per kilo base pairs. The solid green line is a fit to the mutation rate data with the same function as in *Equation 13*. The fit parameters are $\bar{v} = 2.4\,\text{kbp}^{-1}\text{gen}^{-1}$, $\delta = 0.18$, $\omega = 4.9\,\text{rad}\,\text{Mbp}^{-1}$ and $\phi = 1.93\,\text{rad}$. (**b**) Temperature dependence of angular frequency of oscillation $\omega$ (squares). (**c**) phase $\phi$ (squares). Green triangles in (**b**) and (**c**) represent the angular frequency and phase, respectively, from the fit to the mutation rate data with *Equation 13*. (**d**) Diffusion coefficient $D$. Circles represent individual fitted values of diffusion coefficients. Blue circles represent cases in which the fitted value of $D$ is either zero or not significant (see SI). This occurs in two out of nine cases for 37°C and three out of nine cases for each of the other temperatures.

The online version of this article includes the following figure supplement(s) for figure 4:

**Figure supplement 1.** Alternative version of *Figure 4a*.

**Figure supplement 2.** Oscillatory model with and without diffusion.

Our results show that the diffusion coefficient $D$ is quite small. For about one third of our experimental realizations at each temperature, our fitted value of $D$ is not significant according to the Akaike information criterion (see *Figure 4d* and *Figure 4—figure supplement 2*). For comparison, we estimate the equilibrium diffusion constant of replisomes in the cytoplasm from the Stokes-Einstein relation as $D_{\text{SE}} \approx 6\,\text{kbp}^2\text{s}^{-1}$ (see Appendix 5), of the same order of magnitude as our fitted values, see *Table 1* and *Figure 4*. These results suggest that, despite their high average speed, the fluctuations of the replisome position are remarkably similar to the equilibrium case.

The diffusion coefficient determines the uncertainty about the genome site where the two replisomes meet. In the absence of diffusion ($D = 0$), replisomes would always meet at the midpoint on the circular genome. For $D > 0$, we estimate the typical size $l_D$ of the region in which the two replisomes meet as follows. Since the fitted values of $\delta$ and $D$ are both small, we approximate the replication time as $\tau_C \approx L/(2\bar{v})$. In this time, the accumulated uncertainty due to diffusion is equal to $l_D \approx 2\sqrt{2D\tau_D}$. From our estimated diffusion coefficients and average velocities, we obtain values of $l_D$ on the order of $100 - 200$ kbp, depending on temperature.

We remark that our bacterial cultures are grown in LB medium. Although the growth curves appear exponential before saturating (see Methods), the nutrient composition can be such that the assumption of steady growth made in our model is not valid. It is therefore important to scrutinize whether the oscillations can be a consequence of this issue. To this aim, we analyzed a version of the model in which the fork firing rate is not steady, but gradually declines with time, see Appendix 7. We find that the discrepancy between the DNA abundance distributions predicted by this model and by the steady-state model is very small compared to the oscillations we observe. This observation supports that a potential lack of steady-state in the LB medium is not a likely cause for the oscillations.

## Discussion

In this paper, we infer the dynamics of replisomes from the DNA abundance distribution in a growing bacterial population. Our theory can be seen as a generalization of the classic Cooper-Helmstetter theory (*Cooper and Helmstetter, 1968*; *Bremer and Churchward, 1977*), that permits to estimate the duration of the replication period from the abundance of certain genomic locations in a growing population, see e.g. (*Zheng et al., 2016*; *Si et al., 2017*). While the Cooper-Helmstetter theory assumes constant replisome speed, our approach allows for varying speeds. We test our method by measuring the DNA abundance distribution of *E. coli* populations growing at different temperatures. We thereby accurately estimate the average speed of replisomes in vivo, and their speed variations along the genome.

We find that the dependence of the average replisome speed on the temperature is well described by an Arrhenius law, similar to that governing the population growth rate. This quantitative dependence can be used to deduce other laws governing bacterial physiology at varying temperature. For example, we argue that precise DNA-protein homeostasis requires the cell size to mildly vary with temperature. This prediction is in qualitative agreement with previous observations (*Shehata and Marr, 1975*; *Trueba et al., 1982*) and calls for more systematic measurements of cell parameters at varying temperature, similar to what has been done in the case of varying nutrient composition (*Si et al., 2017*).

Our approach reveals a wave-like oscillation of the replisome speed along the *E. coli* genome. The relative amplitude of these oscillations ranges from 10% to 20% of the average replisome speed. A quantitatively similar pattern was observed in the bacterial mutation rate along the DNA of an *E. coli* mutant strain (*Niccum et al., 2019*) and of other bacterial species (*Dillon et al., 2018*). This similarity suggests that the two phenomena have a common dynamical origin. In particular, we hypothesize that this correlation could be a manifestation of the trade-off between accuracy and speed that characterizes DNA polymerases (*Sartori and Pigolotti, 2013*; *Banerjee et al., 2017*; *Fitzsimmons et al., 2018*). Because of this trade-off, any mechanism increasing the speed of a polymerase is expected to increase its error rate as well.

Our analysis of the frequency of these oscillations supports that this pattern may originate from a process synchronized with the cell cycle (*Dillon et al., 2018*), whose activity alters the replisome function. An alternative hypothesis is that the oscillations originate from competition among replisomes for shared resources, such as nucleotides. According to this idea, the firing of new forks can hinder the progression of existing replisomes. The frequency of oscillations is compatible with both explanations. The following additional evidence supports the latter hypothesis. We did not observe appreciable oscillations for our lowest temperature of 17°C, which according to our estimates falls outside the multi-fork replication regime. Further, we found that oscillations disappeared when analyzing previous data from a culture grown in a minimal medium (*Midgley-Smith et al., 2018*), where multi-fork replication is also not expected. On one hand, these facts point to competition between multiple replisomes in the same cell as a likely source for the oscillations. On the other hand, given the difficulty of

obtaining steady exponential growth in LB medium, further experiments will be important to assess an eventual effect of the growth medium on the DNA abundance shape.

Beside these regular and repeatable variations, our analysis shows that random fluctuations of replisome speed are quite small, leading to an uncertainty of about $100 - 200\,\text{kbp}$ on the location of the replisome meeting point. In bacteria, the terminal region of replication is flanked by two groups of termination (Ter) sites having opposite orientations. Ter sites are the binding sites for the Tus protein and permit passage of replication forks in one direction only (**Elshenawy et al., 2015**), so that the two groups effectively trap the two forks in the terminal region (**Duggin et al., 2008**). Out of the ten Ter sequences in *E. coli*, only two of them (TerB and TerC) are within $100 - 200\,\text{kbp}$ of the point diametrically opposite to the origin. These two sequences have the same orientation. Our result therefore implies that most Ter sequences are usually not needed to localize the replisome meeting point. This prediction is consistent with previous observations that the phenotypes of Tus- *E. coli* mutants (**Roecklein et al., 1991**) or mutants lacking Ter sequences (**Duggin et al., 2008**) do not appear distinct from that of the wild type.

Quantitative modeling of the DNA abundance distribution has the potential to shed light on aspects of replisome dynamics beyond those explored in this paper. For example, it was observed that the knockout of proteins involved in the completion of DNA replication leads to either over-expression or under-expression of DNA in the terminal region (**Wendel et al., 2014**; **Wendel et al., 2018**; **Sinha et al., 2018**). Incorporating the role of these proteins into our model will permit to validate possible explanations for these patterns. More in general, our approach is simple and general enough to be readily applied to other bacterial species, to unravel common principles and differences in their DNA replication dynamics.

## Methods
### Cultivation and DNA extraction
*E. coli* MG1655 was cultured in LB medium supplemented with 50 mM MOPS pH 7.2 and 0.2% glucose. Overnight cultures grown at 37 °C were diluted into fresh medium and grown until reaching an OD600 of about 1.0 at the target temperature. These cultures were used to inoculate 50 ml medium at the desired temperature in 500 ml Erlenmeyer flasks with baffles at a target OD of 0.01. Cultivation was performed with shaking at 250 rpm. OD was determined with a NanoDrop One in cuvette mode.

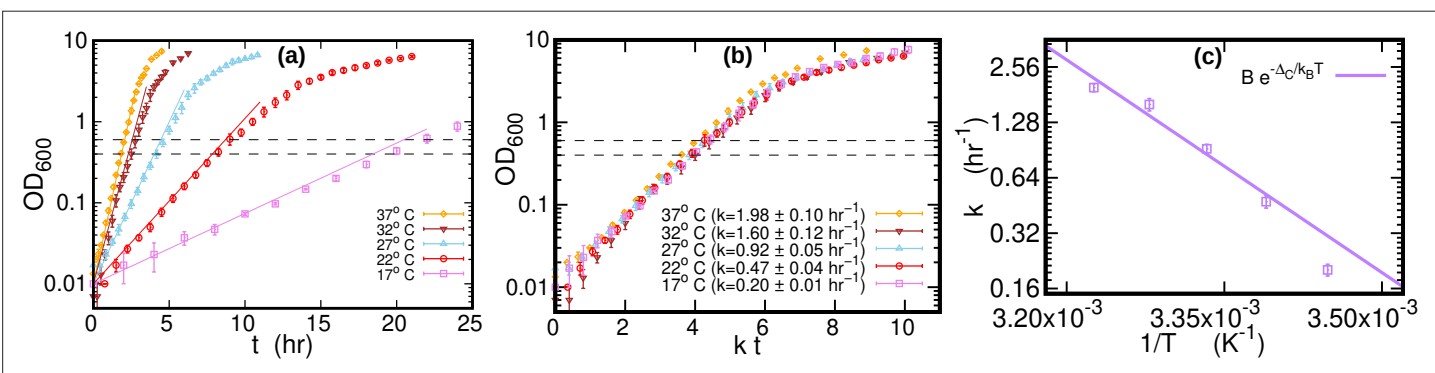

**Figure 5.** Growth curves at different temperatures. (**a**) Optical density (OD) as a function of time at different temperatures. Each curve is averaged over three different replicates at the same temperature. Error bars represent standard deviations. Dashed lines mark the OD window in which the cells are harvested. Solid lines represents the exponential growth curve for each temperature. We computed the growth rate $k_i$ for each sample $i = 1, 2, 3$ at a given temperature by fitting the optical density to a logistic function $a_i/[1 + b_i \exp(-k_i t)]$, where $a_i$ and $b_i$ are sample-specific constants (**Zwietering et al., 1990**). The growth rate $k$ for each temperature is the average of the $k_i$s. (**b**) Same data as in (**a**), but the time in the x-axis is scaled by the growth rate $k$ at each temperature. As a result of this rescaling, the growth curves collapse on each other. (**c**) Average growth rate as a function of temperature. The solid purple line is an Arrhenius fit to the data (see **Figure 2**), resulting in $B = (6.0 \pm 24.9) \times 10^{12}\,\text{hr}^{-1}$ and $\Delta_C = (74 \pm 10)\,\text{kJmol}^{-1}$. We exclude the data point for $T = 17^\text{o}\text{C}$ from the fit.

The online version of this article includes the following figure supplement(s) for figure 5:

**Figure supplement 1.** Individual growth curves at different temperatures.

The growth curves of *E. coli* were highly repeatable (over three replicate experiments for each temperature), see *Figure 5a* and *Figure 5b*. We computed the growth rate at each temperature by fitting a logistic function to individual growth curves, see *Figure 5c*. When the time was rescaled by the average growth rate, OD of different temperatures collapsed along a single curve. The cultures (1.4 ml) were harvested by centrifugation at 21,000 g for 20 s after reaching an OD of around 0.5 (mid-exponential phase, dashed lines in *Figure 5a*). Cells were kept growing for at least 45 doubling times (as measured in exponential phase) to reach stationary phase. Samples of 0.2 ml from the stationary phase cultures grown at 17°C, 27°C, and 37°C were harvested for DNA extraction. The pellets were immediately frozen at –80 °C until DNA extraction. DNA was extracted in parallel using Genomic DNA Purification Kit from Thermo Fisher Scientific.

## Sequencing

We sequenced three samples in the exponential phase from different experimental realizations for each temperature. In addition, we sequenced three stationary samples at three different temperatures. DNA samples were sheared by ultrasound using Covaris AFA technology. Libraries were then prepared using the PCR-free NEBNext Ultra II DNA Library Prep Kit for Illumina. Sequencing was performed on a Novaseq6000 using paired-end 150 bp reads.

## Alignment and bias elimination

We aligned reads from each sample using Bowtie2 2.3.4.1 (*Langmead and Salzberg, 2012*), using the MG1655 genome as a reference. We calculated the frequency of reads as a function of the genome coordinate with bin size 10kbp. To attenuate bias, we divided the frequency at each genome coordinate in a sample from the exponential phase by the frequency of the corresponding bin in a stationary sample (*Wendel et al., 2014*; *Midgley-Smith et al., 2018*). We alternatively used all of our three stationary samples to correct the bias of each sample in the exponential phase. Therefore, after bias elimination, we effectively have $3 \times 3 = 9$ different DNA abundance curves in the exponential phase at each temperature. See *Figure 2—figure supplement 1*, *Figure 2—figure supplement 2* and *Figure 2—figure supplement 3* for details.

## Stationary distribution of replisome positions

In this section, we discuss how to compute the stationary distribution of incomplete replisome positions $p^{\text{st}}(x_1, x_2; t)$. We call $n_S(x_1, x_2; t)$ the number density of incomplete genomes at time $t$ with replisome positions at $x_1$ and $x_2$. By definition $\int_0^L dx_1 \int_0^{L-x_1} dx_2 n_S(x_1, x_2; t) = N_S(t)$. It follows from *Equation 7* that this number density evolves according to

$$\frac{\partial}{\partial t} n_S(x_1, x_2; t) = -\vec{\nabla} \cdot [\vec{v} n_S] + D\nabla^2 n_S, \tag{14}$$

where $\vec{\nabla} = (\partial/\partial_{x_1}, \partial/\partial_{x_2})$ and $\vec{v} = (v(x_1), v(x_2))$. We now introduce the normalized probability $p(x_1, x_2; t) = n_S(x_1, x_2; t)/N_S(t)$. By substituting this definition into *Equation 14*, we obtain

$$\frac{\partial}{\partial t} p(x_1, x_2; t) = -\vec{\nabla} \cdot [\vec{v} p] + D\nabla^2 p - kp. \tag{15}$$

The stationary distribution $p^{\text{st}}(x_1, x_2)$ is a time-independent solution of *Equation 15*, see *Equation 8*.

Because of replication completion, the line $x_1 + x_2 = L$ is an absorbing state for the dynamics described by *Equation 15*. *Equation 15* must be consistent with *Equation 2*, which describes the dynamics of incomplete genomes regardless of the coordinates of their replisomes. This implies that the rate β at which replication completes (see *Equation 2*) must equal to the probability flux through the absorbing boundary:

$$\beta = \int_{x_1+x_2=L} \vec{J} \cdot \hat{n} \, dl, \tag{16}$$

where we introduce the probability current $\vec{J}(x_1, x_2) = \vec{v} p - D\vec{\nabla} p$, the unit vector $\hat{n} = (1/\sqrt{2}, 1/\sqrt{2})$, and the infinitesimal line increment $dl$ along the absorbing boundary. Similarly, the probability flux

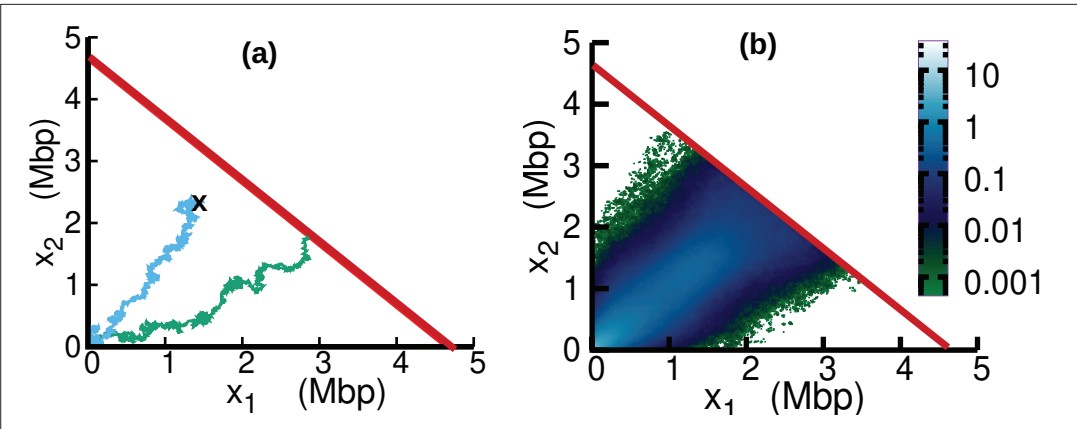

**Figure 6.** Replisome dynamics in the $(x_1, x_2)$ plane. (**a**) Two different trajectories demonstrate two different types of resetting events in our simulations. Trajectories are reset to $x_1 = 0, x_2 = 0$ when the two replisomes complete replication (green trajectory) at the absorbing boundary (solid red line). Additionally, trajectories can be reset from any position to the origin at a rate $k$ (sky blue) to take care of the dilution term in **Equation 15**. (**b**) Replisome position distribution $p^{\text{st}}(x_1, x_2)$ in the steady state. In both panels, parameters are $\bar{v} = 973 \text{bp s}^{-1}$, $\delta = 0.19$, $\omega = 4 \text{rad Mbp}^{-1}$, $\phi = 3.1 \text{rad}$ and $D = 55 \text{kbp}^2 \text{s}^{-1}$. These parameters are on the order of those fitted from experiments (see **Table 1**), except for $D$ which is chosen to be larger for illustration purposes.

entering the system at $(x_1, x_2) = (0, 0)$ must match the rate of replication initiation as given by **Equation 2**.

Given a hypothesis on the speed function $v(x)$ and the diffusion coefficient $D$, we solve **Equation 15** at stationarity using the experimentally measured growth rate $k$. From the stationary solution $p^{\text{st}}(x_1, x_2)$, we obtain β using **Equation 16**. Our approach does not permit to determine the cell division rate α appearing in **Equation 1**-Equation 3. However, this rate is not necessary to compute the DNA abundance distribution, which is expressed by **Equation 9** and **Equation 10** in terms of $p^{\text{st}}(x_1, x_2)$ and β only.

## Average genome length

To compute the average genome length, we first note that the integral of $\mathcal{P}(y)$ is equal to the average genome length $\ell$ in the population

$$\ell = \int_{-L/2}^{L/2} \mathcal{P}(y) dy. \tag{17}$$

Combining **Equation 17**, **Equation 10**, and the fact that $\mathcal{P}(0) = 1$, we obtain a simple relation between the DNA abundance distribution and the average genome length:

$$\ell = \mathcal{A}(0)^{-1}. \tag{18}$$

## Average number of replisomes per complete genome

We estimate the average number of replisomes per complete genome $\mathcal{N}$ in two alternative ways. On the one hand, using **Equation 4**-Equation 6 we find that

$$\mathcal{N} = \frac{2N_S}{N_P + N_T} = \frac{2k}{\beta}. \tag{19}$$

On the other hand, it can be seen in **Figure 2d** that the fraction of complete genome in the population is equal to the ratio $\mathcal{A}(L/2)/\mathcal{A}(0)$ between the DNA abundance at the terminal and at the origin. It follows that

$$\mathcal{N} = \frac{2[\mathcal{A}(0) - \mathcal{A}(L/2)]}{\mathcal{A}(L/2)}. \tag{20}$$

## Constant speed

We focus on the scenario with constant speed and $D = 0$. In this case, the steady solution of *Equation 15* is given by

$$p^{\text{st}}(x_1, x_2) = \frac{ke^{-\frac{k}{2\bar{v}}(x_1+x_2)}}{\bar{v}\left(1 - e^{-kL/(2\bar{v})}\right)}\delta(x_1 - x_2). \tag{21}$$

The rate at which replication completes is equal to

$$\beta = \frac{ke^{-kL/(2\bar{v})}}{1 - e^{-kL/(2\bar{v})}}. \tag{22}$$

Substituting *Equation 21* and *Equation 22* into *Equation 9*, we obtain.

$$\mathcal{P}(y) = e^{-k|y|/\bar{v}}, \tag{23}$$

from which *Equation 11* follows by normalizing, see *Equation 10*.

We exactly solved *Equation 15* also in the case where the speed depends on the genome coordinate, provided that the diffusion coefficient vanishes, see Appendix 6.

## Stochastic simulations

In the case of oscillating speed and $D > 0$, we compute the stationary solution of *Equation 15* using numerical simulations. To this aim, we interpret *Equation 15* as describing a stochastic process subject to stochastic resetting (*Evans and Majumdar, 2011*). Specifically, we perform stochastic simulations of *Equation 7*. In addition to the dynamics described by *Equation 7*, with a stochastic rate equal to the fork firing rate $k$, trajectories are reset to the origin, $x_1 = x_2 = 0$ (blue trajectory in *Figure 6a*). Since the boundary $x_1 + x_2 = L$ is an absorbing state, trajectories that reach this boundary are also reset to the origin (green trajectory in *Figure 6a*). The probability distribution associated with this dynamics evolves according to *Equation 15*. We simulate this stochastic dynamics to estimate the stationary distribution $p^{\text{st}}(x_1, x_2)$ in a computationally efficient way, see *Figure 6b*. We estimate from the same simulations the parameter β as the empirical rate at which the absorbing boundary is reached, see *Equation 16*.

## Acknowledgements

We are grateful for the help and support provided by the Scientific Computing and Data Analysis section of the Research Support Division at OIST. We thank the DNA Sequencing Section of OIST, in particular N Arasaki, for support with sequencing. This work was supported by JSPS KAKENHI Grant No. JP18K03473 (to SP) and a grant from Deutsche Forschungsgemeinschaft (DFG, grant number: 452628014 to SH). We thank M Cencini and P Sartori for feedback on a preliminary version of our manuscript.

## Additional information

### Funding

| Funder | Grant reference number | Author |
| --- | --- | --- |
| Japan Society for the Promotion of Science | JP18K03473 | Simone Pigolotti |
| Deutsche Forschungsgemeinschaft | 452628014 | Samuel Hauf |
| Deutsche Forschungsgemeinschaft | HA9374/1-1 | Samuel Hauf |

The funders had no role in study design, data collection and interpretation, or the decision to submit the work for publication.

## Author contributions
Deepak Bhat, Simone Pigolotti, Conceptualization, Formal analysis, Methodology, Writing – original draft; Samuel Hauf, Performed the experiments; Charles Plessy, Analysis of sequencing data; Yohei Yokobayashi, Performed the experiments

## Author ORCIDs
Deepak Bhat ![ORCID] http://orcid.org/0000-0001-9387-4951
Samuel Hauf ![ORCID] http://orcid.org/0000-0002-3034-8441
Charles Plessy ![ORCID] http://orcid.org/0000-0001-7410-6295
Yohei Yokobayashi ![ORCID] http://orcid.org/0000-0002-2417-1934
Simone Pigolotti ![ORCID] http://orcid.org/0000-0002-6157-6906

## Decision letter and Author response
Decision letter https://doi.org/10.7554/eLife.75884.sa1
Author response https://doi.org/10.7554/eLife.75884.sa2

---

# Additional files

## Supplementary files
• Transparent reporting form

## Data availability
Sequence reads were deposited in the NCBI Sequence Read Archive with links to BioProject accession number PRJNA772106. Corresponding read frequencies along the genome were deposited in Zenodo (DOI:https://doi.org/10.5281/zenodo.5577986).

The following datasets were generated:

| Author(s) | Year | Dataset title | Dataset URL | Database and Identifier |
|---|---|---|---|---|
| Pigolotti S, Bhat D, Hauf S, Plessy C, Yokobayashi Y | 2022 | *Escherichia coli* DNA replication study | https://www.ncbi.nlm.nih.gov/bioproject/?term=PRJNA772106 | NCBI BioProject, PRJNA772106 |
| Pigolotti S, Bhat D, Hauf S, Plessy C, Yokobayashi Y | 2022 | *Escherichia coli* DNA replication study: processed alignment data | https://doi.org/10.5281/zenodo.5577986 | Zenodo, 10.5281/zenodo.5577986 |

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

# Appendix 1

## Age-dependent dynamics of genome types

In this Appendix, we generalize the model embodied in *Equations 1–3* to the more realistic case in which genomes transition from one stage to another with an age-dependent rate. Specifically, we call $\tau$ the time since a genome fired its last fork and define the time-dependent fork firing rate $k(\tau)$. The time-dependent fork firing rate can be expressed in terms of the fork-firing time-distribution $f(\tau)$ as $k(\tau) = f(\tau)/[1 - \int_0^\tau f(\tau')d\tau']$.

We also define the age $a$ of genomes in synthesizing or post-replication stage, i.e. the time they spent in their stage. We define the age-dependent rate of completion $\beta(a)$ and the age-dependent duration of post-replication stage $\alpha(a)$. These rates can be expressed in terms of the distribution of time to complete the synthesizing stage, $g(a)$ and the distribution of time to complete the post-replication stage, $h(a)$. The relations are: $\beta(a) = g(a)/[1 - \int_0^a g(a')da']$ and $\alpha(a) = h(a)/[1 - \int_0^a h(a')da']$.

We call $n_T(\tau; t)$ the number density of templates at time $t$ that fired their last fork at time $t - \tau$. We call $n_S(a, \tau; t)$, and $n_P(a, \tau; t)$ the number density of synthesizing and post-replication genomes (respectively) at time $t$ that fired their last fork at time $t - \tau$ and with age $a$. These densities evolve according to the coupled equations:

$$\frac{\partial n_T}{\partial t} = -\frac{\partial n_T}{\partial \tau} - k(\tau)n_T + \delta(\tau)\int_0^\infty d\tau' \, k(\tau')n_T(\tau'; t) + \int_0^\infty \alpha(a)n_P(a, \tau; t)da \tag{24}$$

$$\frac{\partial n_S}{\partial t} = -\frac{\partial n_S}{\partial a} - \frac{\partial n_S}{\partial \tau} - k(\tau)n_S + \delta(\tau)\int_0^\infty d\tau'k(\tau')\left[\delta(a)n(\tau'; t) + n_S(a, \tau'; t)\right] - \beta(a)n_S \tag{25}$$

$$\frac{\partial n_P}{\partial t} = -\frac{\partial n_P}{\partial a} - \frac{\partial n_P}{\partial \tau} - k(\tau)n_P + \delta(\tau)\int_0^\infty d\tau' \, k(\tau')n_P(a, \tau'; t) + \delta(a)\int_0^\infty \beta(a')n_S(a', \tau; t)da' - \alpha(a)n_P, \tag{26}$$

where $n(\tau; t) = n_T(\tau; t) + \int_0^\infty n_S(a, \tau; t)\, da + \int_0^\infty n_P(a, \tau; t)\, da$ is the total number density of genomes at time $t$ that fired their last fork at time $t - \tau$.

The total numbers of templates, synthesizing, and post-replication genomes are respectively given by $N_T(t) = \int_0^\infty n_T(\tau; t)d\tau$, $N_S(t) = \iint_0^\infty n_S(a, \tau; t)dad\tau$, and $N_P(t) = \iint_0^\infty n_P(a, \tau; t)dad\tau$. Integrating *Equation 24* with respect to $\tau$ and *Equations 25; 26* with respect to $a$ and $\tau$ we obtain

$$\frac{dN_T}{dt} = \iint_0^\infty \alpha(a)n_P(a, \tau; t)dad\tau \tag{27}$$

$$\frac{dN_S}{dt} = \int_0^\infty d\tau' \, k(\tau')n(\tau'; t) - \iint_0^\infty \beta(a)n_S(a, \tau; t)dad\tau \tag{28}$$

$$\frac{dN_P}{dt} = \iint_0^\infty \beta(a)n_S(a, \tau; t)dad\tau - \iint_0^\infty \alpha(a)n_P(a, \tau; t)dad\tau . \tag{29}$$

The total number of genomes, $N(t) = N_T + N_S + N_P$ grows with time as

$$\frac{dN}{dt} = \int_0^\infty d\tau \, k(\tau)n(\tau; t) . \tag{30}$$

If the fork firing rate is independent of $\tau$, $k(\tau) = k$, then the number of genomes grows as $N = N(0)e^{kt}$. A computation of the growth exponent for general case requires knowledge of $n(\tau, t)$. Using the definition of $n(\tau, t)$ in *Equation 24*-Equation 26 leads to

$$\frac{\partial}{\partial t}n(\tau; t) = -\frac{\partial}{\partial \tau}n(\tau; t) - k(\tau)n(\tau; t) + 2\delta(\tau)\int_0^\infty d\tau' \, k(\tau')n(\tau'; t) . \tag{31}$$

We assume that the age dependent genome population scales with the total number of genomes, $n(t; \tau) = q(\tau; t)N(t)$. This assumption for $n(t; \tau)$ yields an exponential growth for the number of genomes, $N = N(0)e^{\hat{k}(t)t}$, with a time-dependent exponent,

$$\hat{k}(t) = \int_0^\infty k(\tau)q(\tau;t)d\tau\,. \tag{32}$$

Substituting this along with the relation $n(\tau;t) = q(\tau;t)N(t)$ in *Equation 31*, we find

$$\frac{\partial}{\partial t}q(\tau;t) + \hat{k}(t)q(\tau;t) = -\frac{\partial}{\partial \tau}q(\tau;t) - k(\tau)q(\tau;t) + 2\delta(\tau)\int_0^\infty d\tau'\, k(\tau')q(\tau';t)\,. \tag{33}$$

In the steady state, $q_{st}(\tau) \equiv q(\tau;t \to \infty)$ is independent of time. In this limit, the growth rate in *Equation 32* is also time-independent, $\hat{k} = \int_0^\infty k(\tau)q_{st}(\tau)d\tau$. Using these conditions in *Equation 33*, we find

$$q_{st}(\tau) = 2\hat{k}e^{-\hat{k}\tau}\left[1 - \int_0^\tau f(\tau')d\tau'\right]\,, \tag{34}$$

where we used $e^{-\int_0^\tau k(\tau')d\tau'} = 1 - \int_0^\tau f(\tau')d\tau'$. The normalization condition, $\int_0^\infty q_{st}(\tau)d\tau = 1$, yields the Euler-Lotka equation for the relation between $\hat{k}$ and fork firing time distribution $f(\tau)$,

$$2\int_0^\infty e^{-\hat{k}\tau}f(\tau)d\tau = 1\,. \tag{35}$$

In the steady state, we assume that $n_T(\tau;t) = N_T(t)q_T(\tau)$, $n_S(a,\tau;t) = N_S(t)q_S(a,\tau)$ and $n_P(a,\tau;t) = N_P(t)q_P(a,\tau)$, see, e.g., (*Jafarpour et al., 2018*). Because of the definitions of $N_T$, $N_S$ and $N_P$, we have the normalization conditions

$$\int_0^\infty q_T(\tau)d\tau = 1\,, \qquad \iint_0^\infty q_S(a,\tau)dad\tau = 1\,, \qquad \iint_0^\infty q_P(a,\tau)dad\tau = 1\,. \tag{36}$$

These conditions permit to express the dynamics of genome types as

$$\frac{dN_T}{dt} = \hat{\alpha}N_P\,, \qquad \frac{dN_S}{dt} = \hat{k}N - \hat{\beta}N_S\,, \qquad \frac{dN_P}{dt} = \hat{\beta}N_S - \hat{\alpha}N_P\,. \tag{37}$$

where $\hat{\alpha} = \iint_0^\infty \alpha(a)q_P(a,\tau)dad\tau$ and $\hat{\beta} = \iint_0^\infty \beta(a)q_S(a,\tau)dad\tau$. From *Equation 37*, we obtain in the long time limit:

$$\frac{N_T}{N} = \frac{\hat{\alpha}\hat{\beta}}{(\hat{k}+\hat{\beta})(\hat{k}+\hat{\alpha})}\,, \qquad \frac{N_S}{N} = \frac{\hat{k}}{\hat{k}+\hat{\beta}}\,, \qquad \frac{N_P}{N} = \frac{\hat{\beta}\hat{k}}{(\hat{k}+\hat{\beta})(\hat{k}+\hat{\alpha})}\,. \tag{38}$$

These relations are equivalent to *Equation 4*-Equation 6 for age dependent rates.

To compute $\hat{\beta}$ and $\hat{\alpha}$ more explicitly from *Equation 25* and *Equation 26*, we solve for the marginals $\bar{q}_S(a) = \int_0^\infty q_S(a,\tau)d\tau$ and $\bar{q}_P(a) = \int_0^\infty q_P(a,\tau)d\tau$ respectively. We obtain

$$\bar{q}_S(a) = (\hat{k}+\hat{\beta})e^{-\hat{k}a-\int_0^a \beta(a')da'}\,, \qquad \bar{q}_P(a) = (\hat{k}+\hat{\alpha})e^{-\hat{k}a-\int_0^a \alpha(a')da'}\,. \tag{39}$$

Using these marginals in the definition of $\hat{\beta}$ and $\hat{\alpha}$, we find

$$\hat{\beta} = \frac{\hat{k}\int_0^\infty e^{-\hat{k}a}g(a)da}{1 - \int_0^\infty e^{-\hat{k}a}g(a)da}\,, \qquad \hat{\alpha} = \frac{\hat{k}\int_0^\infty e^{-\hat{k}a}h(a)da}{1 - \int_0^\infty e^{-\hat{k}a}h(a)da} \tag{40}$$

In deriving *Equation 40*, we also used the relations $e^{-\int_0^a \beta(a')da'} = 1 - \int_0^a g(a')da'$ and $e^{-\int_0^a \alpha(a')da'} = 1 - \int_0^a h(a')da'$. In the limiting case of age-independent rates α and β, then $g(a) = \beta e^{-\beta a}$ and $h(a) = \alpha e^{-\alpha a}$, and therefore $\hat{\beta} = \beta$ and $\hat{\alpha} = \alpha$ as expected.

## Appendix 2

### Parameter estimation based on maximum likelihood

In this section, we outline the maximum likelihood method to fit the parameters of the model. We make a histogram of our data with bin size $b = 10$ kbp. We call $B$ the total number of bins. We empirically estimate the DNA abundance in each bin $i$ as

$$\mathcal{E}_i = \frac{N_i^e/N_i^s}{b \sum_{i=1}^{B} (N_i^e/N_i^s)} \, , \tag{41}$$

where $N_i^e$ and $N_i^s$ are the numbers of reads in the $i$ th bin from the exponential and stationary culture, respectively. We assume that the number of reads in each bin follows a Poisson distribution (**Aird et al., 2011**). In the limit of large $N_i^e$ and $N_i^s$, the distribution of the empirical DNA abundance $\mathcal{E}_i$ is Gaussian with a standard deviation

$$\sigma_i = \mathcal{E}_i \sqrt{\frac{1}{N_i^e} + \frac{1}{N_i^s}} . \tag{42}$$

The assumption of large number of reads is well satisfied for our sequencing depth: $N_i^e$ ranges from $2.8 \times 10^4$ to $19.9 \times 10^4$ and $N_i^s$ from $4.6 \times 10^4$ to $9.0 \times 10^4$. We call $\mathcal{A}_i \equiv \mathcal{A}(ib)$ the DNA abundance predicted by our model at bin $i$ for a given set of parameters. The joint likelihood of the empirical DNA abundance in all the bins is given by

$$\mathcal{L} = \prod_{i=1}^{B} \frac{1}{\sqrt{2\pi\sigma_i^2}} e^{-\frac{(\mathcal{E}_i - \mathcal{A}_i)^2}{2\sigma_i^2}} . \tag{43}$$

We fit the model parameters by maximizing the logarithm of the likelihood $\ln \mathcal{L}$.

In the constant speed case, we have a single fitting parameter $k/\bar{v}$. In the oscillatory model without and with diffusion we have four and five fitting parameters, respectively.

# Appendix 3

## Correspondence with the Cooper-Helmstetter Theory

In this section, we study our model in the case of constant speed, constant duration of the B and D stages, and slow growth condition, so that all stages B, C, and D of the cell cycle are present. Our goal is to prove that, under these additional assumptions, our model predicts the same total DNA abundance as the Cooper-Helmstetter model.

The expression for the total DNA abundance in the constant speed case reads

$$\mathcal{C} = \frac{N}{N_T} \int_{-L/2}^{L/2} \mathcal{P}(x)dx = \frac{2\bar{v}}{k} \frac{k+\alpha}{\alpha}[e^{kL/(2\bar{v})} - 1], \tag{44}$$

see **Equation 12**. The analogous expression for the Cooper-Helmstetter model is

$$\frac{L}{k\tau_C} e^{k\tau_D} \left[ e^{k\tau_C} - 1 \right], \tag{45}$$

where $\tau_C$ is the time taken by replisomes to complete replication on the genome and $\tau_D$ is post-replication period (**Cooper and Helmstetter, 1968**).

Since we assumed constant replisome speed, we have $\tau_C = L/2\bar{v}$. Therefore, proving that **Equation 44** and **Equation 45** are equivalent boils down to showing that $e^{k\tau_D} = (\alpha + k)/\alpha$. We call $\tau_B$ the time spent by a bacterium in stage B. In the CH model the times $\tau_B$, $\tau_C$ are constant. The division time is $\tau_{\text{div}} = \tau_B + \tau_C + \tau_D$. The population growth rate is $k = \ln 2/\tau_{\text{div}}$.

Since the division time is constant, the steady-state age distribution is expressed by

$$p(\tau) = \frac{ke^{-k\tau}}{1 - e^{-k\tau_{\text{div}}}} = 2ke^{-k\tau}, \tag{46}$$

see e.g. **Powell, 1956**

We now express the steady-state fractions $f_B$, $f_C$, $f_D$ of cells in stage B, C, D as

$$f_B = \int_0^{\tau_B} p(\tau)d\tau = 2(1 - e^{-k\tau_B}) = 2 - e^{k(\tau_C+\tau_D)} \tag{47}$$

$$f_C = \int_{\tau_B}^{\tau_B+\tau_C} p(\tau)d\tau = 2e^{-k\tau_B}(1 - e^{-k\tau_C}) = e^{k(\tau_C+\tau_D)} - e^{k\tau_D} \tag{48}$$

$$f_D = \int_{\tau_B+\tau_C}^{\tau_{\text{div}}} p(\tau)d\tau = 2e^{-k(\tau_B+\tau_C)}(1 - e^{-k\tau_D}) = e^{k\tau_D} - 1. \tag{49}$$

The ratio of the number of post replication genomes to the number of templates is equal to the fraction of cells in stage D. Therefore, equating the fraction $f_D$ with $N_P/N_T = k/\alpha$ obtained from **Equation 4** and **Equation 6**, we obtain $e^{k\tau_D} = (\alpha + k)/\alpha$ as expected.

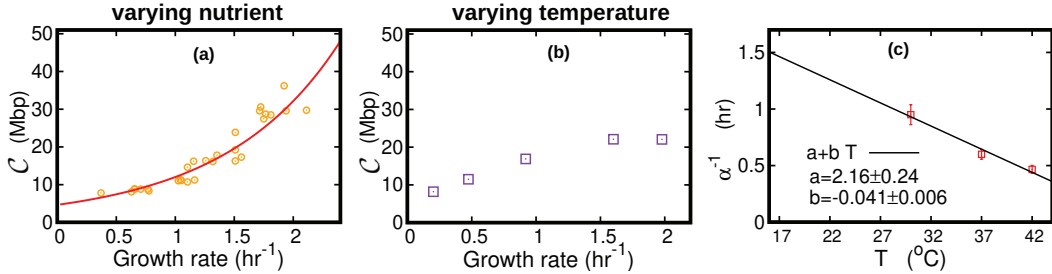

**Appendix 3—figure 1.** DNA content per cell as a function of the growth rate for the case of varying nutrients in (**a**) and of varying temperature in (**b**). In (**a**), the experimental data (orange circles) are from **Si et al., 2017**. The solid red line is from **Equation 45**, in which we used $\tau_C = 38$ min and $\tau_D = 37.1$ min (**Si et al., 2017**). The curve in (**b**) is from **Equation 44**. In this case, we substituted the speed of replisomes and the growth rate of cells at each temperature in **Equation 44**. In addition, we assumed a linear temperature dependence for the post replication duration ($\alpha^{-1}$), see (**c**). The parameters of the linear fit are determined from the data (red squares) reported

in *Stokke et al., 2012* for the LB medium. We used this linear fit to extrapolate the value of α for different temperatures in *Equation 44*.

# Appendix 4

**Appendix 4—table 1.** Comparison between average (over the sample to sample variations) speed estimated in the constant and oscillatory speed models.

Temperature are expressed in Celsius and speeds in $\mathrm{bp\,s}^{-1}$. The last column shows the average growth rate (expressed in $\mathrm{hr}^{-1}$) at different temperatures.

| Temperature | $\bar{v}$(constant speed) | $\bar{v}$(oscillatory speed) | $k$(growth rate) |
|---|---|---|---|
| 17 | 221±17 | 243±36 | 0.20±0.02 |
| 22 | 373±29 | 350±28 | 0.47±0.04 |
| 27 | 528±31 | 542±30 | 0.92±0.05 |
| 32 | 812±64 | 823±65 | 1.60±0.12 |
| 37 | 961±51 | 972±51 | 1.98±0.10 |

## Appendix 5

### Estimate of the diffusion coefficient from the Stokes-Einstein relation

The Stokes-Einstein relation expresses the diffusion coefficient of a spherical particle immersed in a fluid (**Hynes, 1977**). If $r$ is the radius of the particle, $\eta$ and $T$ are viscosity and the temperature of the fluid respectively, then according to the Stokes-Einstein relation, the diffusion coefficient is

$$D_{\text{SE}} = \frac{k_B T}{6\pi\eta r} \tag{50}$$

where $k_B$ is the Boltzmann constant. The radius of an *E. coli* replisome is $r \approx 50\,nm$(**Reyes-Lamothe et al., 2010**) and the viscosity of water is $\eta = 0.7\text{mPa s}$ at room temperature $T = 310\text{K}$. Using that $k_B = 1.38 \times 10^{-23}JK^{-1}$, we estimate a diffusion coefficient of replisomes in water equal to $D_{\text{SE,W}} = 6\mu\text{m}^2\text{s}^{-1}$. The typical base pair distance is $3.4A^o$. Therefore in terms of base-pair (bp), $D_{\text{SE,W}} \approx 56\text{kbp}^2\text{s}^{-1}$. The diffusion constant of large macromolecules in the cytoplasm is found to be about 10 times smaller than in water (**Verkman, 2002**). This results in an estimate of the diffusion constant of replisomes in the cytoplasm of $D_{\text{SE,C}} \approx 6\text{kbp}^2\text{s}^{-1}$, as reported in the Results.

# Appendix 6

## Exact solution of the oscillatory speed case for $D = 0$

In this section, we exactly solve the oscillatory speed model in the absence of diffusion ($D = 0$). For an arbitrary choice of the function $v(x)$ and $D = 0$, the steady state solution of *Equation 15* reads

$$p^{\text{st}}(x_1, x_2) = \delta(x_2 - x_1)\frac{A}{v(x_1)}e^{-\int_0^{x_1} dx'\, k/v(x')} \tag{51}$$

where $A$ is a normalization constant that ensures $\int_0^L dx_1 \int_0^{x_1} dx_2 p^{\text{st}}(x_1, x_2) = 1$. The rate at which replication completes is $\beta = Ae^{-\int_0^{L/2} dx'\, k/v(x')}$. From *Equation 9*, we obtain

$$\mathcal{P}(y) = \frac{A}{k + \beta}e^{-\int_0^{|y|} dy'\, k/v(y')}. \tag{52}$$

For the specific form of $v(x)$ given in *Equation 13*, the integral in *Equation 52* is equal to

$$\int_0^{|y|} dy'\, k/v(y') = \quad \frac{2k}{\bar{v}\omega\sqrt{1-\delta^2}}\left\{ \arctan\left[\sqrt{\frac{1-\delta}{1+\delta}}\tan\left(\frac{\omega|y|+\phi}{2}\right)\right] - \arctan\left[\sqrt{\frac{1-\delta}{1+\delta}}\tan\left(\frac{\phi}{2}\right)\right]\right.$$
$$\left. +\pi\left\lfloor\frac{\omega|y|+\phi+\pi}{2\pi}\right\rfloor - \pi\left\lfloor\frac{\phi+\pi}{2\pi}\right\rfloor\right\}. \tag{53}$$

where $\lfloor\cdot\rfloor$ is the floor function. We use the expression of this integral in *Equation 52* and substitute the result in *Equation 10* to obtain the DNA abundance distribution. We computed the normalization factor of the DNA abundance distribution (i.e., the denominator of *Equation 10*) by numerical integration.

## Appendix 7

### Replisome dynamics with time-dependent fork firing rate

To verify the robustness of our results, we generalize our model to the case in which the fork firing rate $k$ depends on time. For simplicity, we assume that replisomes move with constant speed and without diffusion ($D = 0$).

We assume that the total number of genomes grows with time as

$$N(t) = \theta(t)N_0 e^{\int_0^t \hat{k}(t')dt'} = \theta(t)\frac{a}{1 + b\exp(-kt)} \tag{54}$$

where $N_0 = a/(1 + b)$ is the initial number of genomes and we introduced the time-varying fork firing rate

$$\hat{k}(t) = \theta(t)\frac{bk\exp(-kt)}{1 + b\exp(-kt)} . \tag{55}$$

The step function $\theta(t)$ in **Equation 54** serves to impose that replication occurs for $t \geq 0$ only. The corresponding dynamics of synthesizing genomes reads

$$\frac{\partial}{\partial t}n_S(x_1; t) = \delta(x_1)\hat{k}(t)N(t) - v\frac{\partial n_S}{\partial x_1} , \tag{56}$$

where the delta function ensures that the replication initiates at $x_1 = 0$. As we have set $D = 0$, the coordinate $x_2$ of the second replisome obeys the same dynamics expressed in **Equation 56** and the DNA abundance is symmetric around the replication origin.

The general solution to **Equation 56** is of the form $n_s(x_1; t) = f(x_1 - vt)$. Because of the boundary term at $x_1 = 0$, i.e. $n_s(0; t) = \hat{k}(t)N(t)/v$, the general solution is

$$n_S(x_1; t) = \frac{\hat{k}\left(t - \frac{x_1}{v}\right)N\left(t - \frac{x_1}{v}\right)}{v} . \tag{57}$$

The probability that a randomly chosen genome (either complete or incomplete) includes the genome location $y$ at time $t$ is given by

$$\mathcal{P}(y; t) = \frac{N_P(t) + N_T(t) + \int_{|y|}^{L/2} n_S(x_1, t)dx_1}{N(t)} . \tag{58}$$

where $\int_{|y|}^{L/2} n_S(x_1, t)dx_1$ is the number of incomplete genomes which include the genome location $y$ and $N_P(t) + N_T(t)$ is the total number of complete genomes. From **Equations 54; 55**, and (**Equation 56**) , we obtain

$$\int_{|y|}^{L/2} n_S(x_1; t)dx_1 = N\left(t - \frac{|y|}{v}\right) - N\left(t - \frac{L}{2v}\right) - N_0\theta\left(t - \frac{|y|}{v}\right)\theta\left(\frac{L}{2v} - t\right). \tag{59}$$

Therefore, the total number of incomplete genomes is $N_s(t) = \int_0^{L/2} n_S(x_1; t)dx_1 = N(t) - N\left(t - \frac{L}{2v}\right) - N_0\theta(t)\theta\left(\frac{L}{2v} - t\right)$. Further, because of the conservation of the total number of genomes, the number of complete genomes at time $t$ is

$$N_T + N_P = N(t) - N_s(t) = N\left(t - \frac{L}{2v}\right) + N_0\theta(t)\theta\left(\frac{L}{2v} - t\right). \tag{60}$$

Substituting **Equation 54**, **Equation 59** and **Equation 60** in **Equation 58**, and using **Equation 10**, we compute the DNA abundance distribution for $t > L/(2v)$ as

$$\mathcal{A}(y) = \frac{k/(2v)}{\log\left[b\exp(-kt) + 1\right] - \log\left[b\exp(-kt) + \exp\left(-\frac{kL}{2v}\right)\right]}\frac{1}{1 + b\exp\left[-k\left(t - \frac{|y|}{v}\right)\right]}. \tag{61}$$

We now use this result to test robustness of the steady-state assumption. To this aim, we assume that the number of genomes follows a similar dynamics as the optical density. We accordingly consider, for each sample at each temperature, the parameters $a_i$, $b_i$, $k_i$ obtained from the OD fits (see *Figure 5—figure supplement 1*), and the average speed $\bar{v}$ estimated from the constant speed model (see *Table 1*). We substitute these in *Equation 61* and compute the DNA abundance at time $t = 4/k$. This value is chosen as, in our experiments, cells were extracted after approximately four generations, see *Figure 5*.

We then attempt to fit these DNA abundance distribution with the one predicted the constant speed model, *Equation 11*, by least-square minimization and thus estimating the speed again. The newly estimated speeds at different temperatures are comparable to those inputted in the time-varying fork firing rate model, see *Appendix 7—table 1*. Finally, we plotted the discrepancy ratio in each of the cases, see *Appendix 7—figure 1*. At all temperatures, the discrepancy ratio is much smaller than the oscillations we observed in the data. This result supports that the potential non-stationary nature of the fork firing rate is not likely to be the cause of the oscillations.

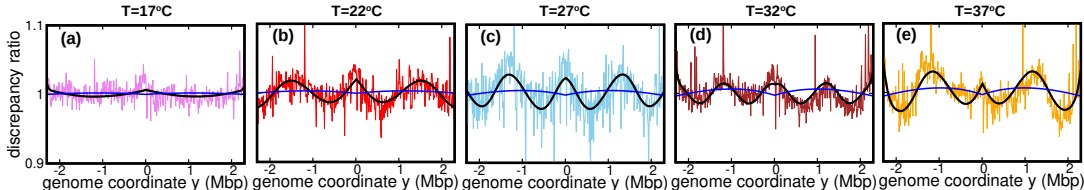

**Appendix 7—figure 1.** Discrepancy ratio (blue line) between DNA abundance computed from the varying fork firing rate model and the constant speed model exhibits significantly smaller variations compared to the discrepancy ratio between experimental DNA abundance and the constant speed model (pink for 17°C , red for 22°C, skyblue for 27°C, brown for 32°C and orange for 37°C). Black line is the prediction of the oscillatory speed model. All plots in this figure are from Sample 1. The effect of relaxing the steady-state assumption is similarly small for the other samples.

We remark that, in the estimate presented in this section, we assumed that cells were not growing before the initial time $t = 0$. In our culture, we expect the effect of saturation to be even milder since cells were inoculated from an already exponentially growing culture.

**Appendix 7—table 1.** The speed inputted to the time varying fork firing rate model to generate the DNA abundance distribution is comparable with the re-estimated speed by fitting the resulting DNA abundance distribution to the constant speed model.
Reported uncertainties in the re-estimated speed represent standard deviations over the replicates.

| Temperature (°C) | Speed inputted to the varying fork firing rate model (bps$^{-1}$) | Re-estimated speed (bps$^{-1}$) |
|---|---|---|
| 17 | 221 | 233±8 |
| 22 | 373 | 393±23 |
| 27 | 528 | 558±25 |
| 32 | 812 | 846±37 |
| 37 | 961 | 1009±48 |

