## [Editor Report]

This manuscript combines theory with experiments to characterize the replication speed of bacteria chromosomes through the cell cycle. The authors show oscillatory patterns in the replication speed of *E. coli*, which they relate to the heterogeneity of mutation rates along the genome, suggesting a tradeoff between the speed and accuracy of replication. This work presents an elegant approach for investigating bacterial growth from a systems biology perspective.

---

## [Decision Letter]

**Decision letter after peer review:**

Thank you for submitting your article "Speed fluctuations of bacterial replisomes" for consideration by *eLife*. Your article has been reviewed by 3 peer reviewers, and the evaluation has been overseen by a Reviewing Editor and Jessica Tyler as the Senior Editor. The following individuals involved in review of your submission have agreed to reveal their identity: Ariel Amir (Reviewer #3).

Essential revisions:

1. The interpretation in terms of a speed-error trade-off is rather speculative and perhaps less emphasis should be placed on it (e.g. in the abstract and the beginning of p.9).

2. There is a possible problem with LB used as growth media. To infer the replication speed, the firing rate needs to be determined, which in the steady state is equal to the growth rate. However, it is close to impossible to achieve well defined steady growth rate when cells grow in LB since nutrients become depleted and growth rate decreases quite substantially. Which growth rate was used to infer the speed (It was unclear) An experiment in a minimal medium (e.g. with glucose as carbon source) with a well-defined growth rate might provide much more reliable results.

3. This study was done in the regime of fast growth. It is known that for *E. coli* there are many changes in the cell cycle properties when the doubling time (at 37 Celcius) exceeds 60 minutes (i.e. the regime where there are no overlapping replication forks). How do the results change in slow growth conditions?

4. Oscillation of replication speeds along the chromosome is reported for different temperatures. This is an interesting observation. While an experimental investigation of these oscillations is beyond the scope of this manuscript, a discussion about the physiological implications of these oscillations would be insightful.

5. The idea of using the frequency inferred from sequencing was also used in: Growth dynamics of gut microbiota in health and disease inferred from single metagenomic samples, Korem et al. Science (2015). Are the oscillations also observed in those measurements? If so, is there information which could be gleaned from them?

6. Figure 2D: Why do replisome numbers increase with increasing temperature and how does this observation reconcile with the increased speeds as well? Do both replication fork numbers and speed of individual replisomes contribute to the observed growth rates across temperatures?

7. There is a systematic difference in the dependence of speed and growth rate on temperature, which the authors discuss. What is the expected change in cell size if the Cooper-Helmstetter model is correct? Should it be observable experimentally? Is it?

8. Please clarify the arguments in Appendix 2 and L140-150.

9. Please modify the Discussion section so it is more focused on the conclusions of this work.

10. Provide a more detailed comparison of the considered approach with previously published work. How is the introduced method different and to what extent it goes beyond the previous work? Is this model more predictive of speed?

11. Lines 131-133: why is the average DNA per cell the product of the two other averages? Is this an approximation or are the two other variables uncorrelated?

*Reviewer #1 (Recommendations for the authors):*

I missed a more detail comparison of the considered approach with previously published approaches and results. How is the introduced method different and to which extent is it going beyond the published results? Does it allow a more precis derivation of speed? What was is with the previous approaches which was limiting the quantification? As presented the introduction of the method is the major contribution of this paper and I thus think that these questions should be clearly answered.

There is a possible problem with LB used as growth media. To infer the replication speed, the firring rate needs to be determined, which in steady state growth is equal to the growth rate. However, it is close to impossible to achieve well defined steady growth rate when cells grow in LB since nutrients become depleted and growth rates decreases quite substantially. Which growth-rate was thus used to infer the speed (It was unclear) An experiment in a minimal medium (e.g. with glucose as carbon source) with a well define growth rate might provide much more reliable results. An observation confirming that (average) replication speeds are not varying much with media conditions (at the same temperature) would also make a further important physiological point.

The authors show that the interfered replication speed is changing strongly with temperature. This has been reported before. With their measurements the authors state that the change is comparable to the change of growth when temperature is varied, but the authors do not comment much about the physiological relevance of this relation. An Arhenius like change of enzymatic reactions is expected for simple enzymatic processes, but that a similar relation holds for the entire growth rate is remarkable and still poorly understood, as emphasized by many studies. Given this context, what do we learn about the temperature dependence of growth by the observation that the replication speed is changing in an Arrhenius like form? The data the authors present show that the change in replication speed differs from the change in growth rate with a steeper change of growth rates. What does this mean? Does this support for example the idea that replication speeds are not so much a bottleneck of cell-growth but replication is efficiently regulated to prevent DNA replication from impeding growth?

From a physiological point of view, I think the oscillation of replication speeds with position along the chromosome are more interesting than the variation of speed with temperature. However, it remains unclear how well such oscillations and their frequency are maintained across conditions and maybe even more robustly a function of growth rate. I think the manuscript would thus largely benefit from the analysis of oscillations when cells grow with different rates but at the same temperature. To keep the number of new experiments small, this could for example just be the quantification for growth in a minimal media condition.

*Reviewer #3 (Recommendations for the authors):*

Feedback for the authors:

1. Figure 2D: Why do replisome numbers increase with increasing temperature and how does this observation reconcile with the increased speeds as well? Do both replication fork numbers and speed of individual replisomes contribute to the observed growth rates across temperatures?

2. Arguments in Appendix 2 and L140-150 are unclear and this reviewer is unable to understand the context of the same.

3. Did the authors consider sequencing a dnaQ proof reading mutant (PMID: 8610131) to test the speed vs accuracy tradeoff hypothesis?

4. The Discussion section needs to be more focused on what the main conclusions of this work are. For example, the discussion on replication termination is not very informative. Instead, it would have been helpful if the authors could have explained how they envision the impact of temperature on replisome speed and cell cycle progression, a relationship that seems distinct from the impact of media/ nutrient on these dynamics. Secondly, do mutation rates also increase with increasing temperature, and do the authors envision mutation hotspots given their periodic positional data on the replisome? If so, what determines this periodicity? In the current manuscript, these ideas are not synthesized in a coherent fashion, that makes it harder to appreciate the unique findings.

[Editors' note: further revisions were suggested prior to acceptance, as described below.]

Thank you for resubmitting your work entitled "Speed variations of bacterial replisomes" for further consideration by *eLife*. Your revised article has been evaluated by Jessica Tyler (Senior Editor) and a Reviewing Editor.

The manuscript has been improved but there are some remaining issues that need to be addressed, as outlined below:

As noted by reviewer #1, the statement that the growth rate is in the steady state in LB is not very solid. Please address the concerns regarding this issue. Unless the authors feel that it is necessary, we do not require additional experiments. However, the authors should:

1. Clarify throughout the manuscript the nuances regarding the lack of stationarity in LB.

2. Improve the growth figures to better show the dependence of growth rate on temperature, and whether the populations are in the exponential growth phase.

3. Further elaborate if the oscillations could be due to the fact that the population may have not reached the steady state in LB.

*Reviewer 1 (Recommendations for the authors):*

I feel the requests and suggestions which were presented to the authors to revise the presented manuscript were very reasonable However, after reading the revised manuscript and the responses provided by the authors I now feel that the authors did not even do the bare minimum to address the raised concerns. This particularly concerns a major concern about steady state growth which I raised, and which have been discussed away very carelessly by the authors. I do not support publication until this issue is honestly addressed and fixed in the manuscript as it relates to the major finding promoted by the authors: oscillating replication speeds at fast growth.

In their reply the authors commented on my concern about the steady growth assumption in LB and state that their ‘medium leads to well-defined steady growth rates’. I think this statement is simply false. The fact that a growth-rate can be reproducibly obtained is not sufficient to prove that cells are in a steady state. For that one needs to show that the same growth-rate is obtained over several generations. Why is being picky on this issue so important? Because the authors assume a steady state in their method to infer replication speeds (with e.g. the firing rate being equal the growth rate etc). But it really remains unclear how well the inference method works in cultures which adjust to changing conditions and which are thus not in a steady state. This is a real concern for growth in LB which the authors chose as major growth condition. It is impossible to obtain a good steadily growing culture in LB as the availability of amino acids and other nutrient sources is drastically changing. This can for example be seen in growth curves where the density clearly derivates from an exponential fit when analyzing growth over several doublings. So with this the question remains: are the oscillations the authors report real or are they rather an artifact of falsely assuming steady state growth when doing their analysis. This concerns is highlighted even further by the newly presented findings stating that oscillations in speed are not observed for slower growth in minimal medium. So are these oscillations only observed at fast growth in LB because oscillations in speed are a biological reality at fast growth, or are they observed because a steady state assumption has been falsely assumed? I really want to be constructive here but feel the authors must address this concern very rigorously before this work should be published. So let me try to phrase my concerns in even more detail and in the context of the author's reply

To address my concern the authors simply showed the growth-curves in Figure 5 and stated that this supports cultures are in exponential steady growth. However, the presented analysis is not sufficient to prove steady state. For example, there are not exponential fits shown which prove that growth-rates are maintained for reasonable ranges of density and time. In fact, when looking at the presented panel 5a it appears that the growth-curves at different temperatures are pretty much comparable, while a more detailed fit presumably used to infer growth-rate values (instantaneous rates at a specific OD/timepoint) suggest that growth-rates are different and this difference is what the authors use in the second panel of that figure and to discuss the relation between growth rate and replication speed. Given what is known about growth on LB a zoomed in version of these growth curves will reveal that growth is not really exponential for a reasonable long amount of time (say dozens of generations until protein levels reach a steady state) and as such no steady state have been obtained. In this context I strongly believe that a more careful experimentation with a well-defined steady states in a well-defined media (e.g. rich defined media with exponential growth and the some growth rates observed over > 10 doublings) would have been way better to establish that the inferred oscillations in speed are a real biological phenomenon. I understand that the authors are reluctant to add such additional experiments and I am not insisting on them. However, the concern needs to be taken seriously. I think the authors should:

1) Clearly state in the manuscript that the presented measurements have not been taken during a well-defined steady state

2) Plot and analyze better the growth-curves such that the growth behavior and the differences in growth for different temperatures become visible). In such a plot exponential fits to the growth curves should also be shown which directly allow the reader to compare how well-defined and distinct the growth-rates really are.

3) Raise in the conclusion of the paper that there is a possibility that oscillations might occur because the cultures grew in LB and have thus not reached steady state.

4) Include some sort of error propagation analysis which shows how strong the errors in replication speed can be when assuming a culture not yet in steady growth but running the analysis to infer steady state with the assumption of a steady state. For example, assuming the number of cells with different replication forks is still slowly adjusting with a certain timescale and assuming the replication speed is constant, use a fully dynamical model to derive a DNA abundance distribution. Then inferring the speed by analysis the distribution (using the presented approach assuming steady state).

---

## [Author Response]

Essential revisions:1. The interpretation in terms of a speed-error trade-off is rather speculative and perhaps less emphasis should be placed on it (e.g. in the abstract and the beginning of p.9).

We agree with the Reviewer that, strictly speaking, this interpretation is speculative, although the degree of correlation between mutation experiments and the speed oscillations makes a rather compelling case. In the revised version, we place less emphasis on this interpretation as requested.

2. There is a possible problem with LB used as growth media. To infer the replication speed, the firing rate needs to be determined, which in the steady state is equal to the growth rate. However, it is close to impossible to achieve well defined steady growth rate when cells grow in LB since nutrients become depleted and growth rate decreases quite substantially. Which growth rate was used to infer the speed (It was unclear) An experiment in a minimal medium (e.g. with glucose as carbon source) with a well-defined growth rate might provide much more reliable results.

The LB medium that we used was treated to guarantee an extended exponential growth regime. The growth curves that we observed at different temperature are highly repeatable and present a clear exponential regime. The resulting estimates of the growth rate *k* are very consistent among replicates, with variations on the order of 5%. Moreover, when rescaled by the estimated doubling time, all growth curves at different temperatures collapse on each other. Taken together, these observations show that our medium leads to well-defined steady growth rates.

We realized that some of these aspects were not clearly presented in the manuscript, partially because we presented the growth curves in a supplementary figure. In the revised version, we present the growth curves in the Methods (Figure 5) and discuss the growth rate estimation in more detail.

Additionally, we present an analysis of *E. coli* growing in a minimal medium using data from a previous reference. We discuss these results in the context of the next comment.

3. This study was done in the regime of fast growth. It is known that for *E. coli* there are many changes in the cell cycle properties when the doubling time (at 37 Celcius) exceeds 60 minutes (i.e. the regime where there are no overlapping replication forks). How do the results change in slow growth conditions?

We thank the Reviewer for this comment. In the revised version, we present an additional analysis of sequencing data from *E. coli* growing in a minimal medium (data from Midgley-Smith et al., 2018), see Figure 3—figure supplement 4. We did not observe appreciable speed oscillations in this case. This result suggests that oscillations are linked with the multiple forks regime and disappear when the cell cycle is slowed down by either reducing temperature or nutrient composition. As discussed in the revised manuscript, this result supports the hypothesis that the cause of the oscillations might be competition among replisomes.

4. Oscillation of replication speeds along the chromosome is reported for different temperatures. This is an interesting observation. While an experimental investigation of these oscillations is beyond the scope of this manuscript, a discussion about the physiological implications of these oscillations would be insightful.

We thank the Reviewer for this comment. In the revised Discussion, we expand on the physiological implications of our results, in particular on the hypothesis that these oscillations are linked with the multiple fork regime of replication. The revised Discussion also expands on other implications of our work for bacterial physiology, in particular on the laws governing DNA-protein homeostasis at varying temperature.

5. The idea of using the frequency inferred from sequencing was also used in: Growth dynamics of gut microbiota in health and disease inferred from single metagenomic samples, Korem et al. Science (2015). Are the oscillations also observed in those measurements? If so, is there information which could be gleaned from them?

Author response image 1

**Author response image 1. sa2fig1:** Comparison between (a) DNA abundance (wild type *E. coli*) reported in Korem et al Science (2015) and (b) our experiments. The data in (b) is for T = 37°C and stationary temperature corresponding to 17°C (used for bias elimination).

We thank the Reviewer for pointing out this interesting reference. Following this suggestion, we reanalyzed the DNA abundance from the *E. coli* sequencing data by Korem et al. We found that this dataset is characterized by a much smaller coverage than our experiment. As a result, the DNA abundance distribution is too noisy to infer replisome speed variations, see Figure 1 in this document. In any case, in the Introduction of the revised version we cite this reference as another important application of the DNA abundance distribution.

6. Figure 2D: Why do replisome numbers increase with increasing temperature and how does this observation reconcile with the increased speeds as well? Do both replication fork numbers and speed of individual replisomes contribute to the observed growth rates across temperatures?

We thank the Reviewer for this comment. The key point is that, at increasing temperature, the average replisome speed does not increase as fast as the growth rate, as it can be seen from Figure 2c in the manuscript.

Therefore, multiple fork replication is necessary to fill this gap. In conclusion, both replication fork numbers and speed of individual replisomes provide a significant contribution, as the Reviewer suggests. We clarify this point in the revised draft.

7. There is a systematic difference in the dependence of speed and growth rate on temperature, which the authors discuss. What is the expected change in cell size if the Cooper-Helmstetter model is correct? Should it be observable experimentally? Is it?

Assuming perfect DNA–protein homeostasis, the expected change in cell size should be proportional to the DNA content as shown in Appendix 2, Figure 1b. We are not aware of recent systematic studies of the dependence of cell size on temperatures. Trueba et al. (1982) suggest a moderate increase of the cell size with the growth rate, which seems compatible with our theory. However, this dependence strongly depends on the choice of the medium and the paper only reports a few data points. A systematic study of this interesting issue would require additional experiments, which are beyond the scope of our work.

In the revised version, we clarify our prediction on the cell size behavior on temperature, and comment more extensively on its implications in the Discussion section.

8. Please clarify the arguments in Appendix 2 and L140-150.

We rephrased the arguments to make them clearer.

9. Please modify the Discussion section so it is more focused on the conclusions of this work.

We thank the reviewers for this suggestion. We substantially revised the Discussion section, focusing on the conclusions of our work.

10. Provide a more detailed comparison of the considered approach with previously published work. How is the introduced method different and to what extent it goes beyond the previous work? Is this model more predictive of speed?

The main novelty of our method is that it permits to characterize speed variation of the replisomes. In this sense, it can be seen as a generalization of the classic Cooper-Helmstetter theory, that instead assumes constant replisome speed. We clarify this point in the first paragraph of the revised Discussion.

11. Lines 131-133: why is the average DNA per cell the product of the two other averages? Is this an approximation or are the two other variables uncorrelated?

We thank the Reviewer for this observation. Our model of genome dynamics embodied in Equations (3, 4, 5) assumes, for simplicity, that genomes evolve independently. Because of this assumption, the two averages factorize. We clarify this point in the revised manuscript.

Reviewer #1 (Recommendations for the authors):I missed a more detail comparison of the considered approach with previously published approaches and results. How is the introduced method different and to which extent is it going beyond the published results? Does it allow a more precis derivation of speed? What was is with the previous approaches which was limiting the quantification? As presented the introduction of the method is the major contribution of this paper and I thus think that these questions should be clearly answered.

The main novelty of our method is that it permits to characterize speed variation of the replisomes. In this sense, it can be seen as a generalization of the classic Cooper-Helmstetter theory, that instead assumes constant replisome speed. We clarify this point in the first paragraph of the revised Discussion.

There is a possible problem with LB used as growth media. To infer the replication speed, the firring rate needs to be determined, which in steady state growth is equal to the growth rate. However, it is close to impossible to achieve well defined steady growth rate when cells grow in LB since nutrients become depleted and growth rates decreases quite substantially. Which growth-rate was thus used to infer the speed (It was unclear) An experiment in a minimal medium (e.g. with glucose as carbon source) with a well define growth rate might provide much more reliable results. An observation confirming that (average) replication speeds are not varying much with media conditions (at the same temperature) would also make a further important physiological point.

The LB medium that we used was treated to guarantee an extended exponential growth regime. The growth curves that we observed at different temperature are highly repeatable and present a clear exponential regime. The resulting estimates of the growth rate *k* are very consistent among replicates, with variations on the order of 5%. Moreover, when rescaled by the estimated doubling time, all growth curves at different temperatures collapse on each other. Taken together, these observations show that our medium leads to well-defined steady growth rates.

We realized that some of these aspects were not clearly presented in the manuscript, partially because we presented the growth curves in a supplementary figure. In the revised version, we present the growth curves in the Methods (Figure 5) and discuss the growth rate estimation in more detail.

Additionally, we present an analysis of *E. coli* growing in a minimal medium using data from a previous reference. We discuss these results in the context of the last Reviewer’s comment.

The authors show that the interfered replication speed is changing strongly with temperature. This has been reported before. With their measurements the authors state that the change is comparable to the change of growth when temperature is varied, but the authors do not comment much about the physiological relevance of this relation. An Arhenius like change of enzymatic reactions is expected for simple enzymatic processes, but that a similar relation holds for the entire growth rate is remarkable and still poorly understood, as emphasized by many studies. Given this context, what do we learn about the temperature dependence of growth by the observation that the replication speed is changing in an Arrhenius like form? The data the authors present show that the change in replication speed differs from the change in growth rate with a steeper change of growth rates. What does this mean? Does this support for example the idea that replication speeds are not so much a bottleneck of cell-growth but replication is efficiently regulated to prevent DNA replication from impeding growth?

We thank the Reviewer for this comment. In the revised version, we expand on the physiological relevance of the Arrhenius dependence of the replisome speed on temperature. In particular, we argue that the different slopes of the growth rate and replisome speed as a function of temperature imply that the average number of forks per cell must vary as a function of temperature as well. Moreover, in the revised Discussion, we comment on how the Arrhenius relation leads to a prediction about the behavior of the cell size as a function of temperature.

From a physiological point of view, I think the oscillation of replication speeds with position along the chromosome are more interesting than the variation of speed with temperature. However, it remains unclear how well such oscillations and their frequency are maintained across conditions and maybe even more robustly a function of growth rate. I think the manuscript would thus largely benefit from the analysis of oscillations when cells grow with different rates but at the same temperature. To keep the number of new experiments small, this could for example just be the quantification for growth in a minimal media condition.

We thank the Reviewer for this comment. In the revised version, we present an additional analysis of sequencing data from *E. coli* growing in a minimal medium (data from Midgley-Smith et al., 2018), see Figure 3—figure supplement 4. We did not observe appreciable speed oscillations in this case. This result suggests that oscillations are linked with the multiple forks regime and disappear when the cell cycle is slowed down by either reducing temperature or nutrient composition. As discussed in the revised manuscript, this result supports the hypothesis that the cause of the oscillations might be competition among replisomes.

Reviewer #3 (Recommendations for the authors):Feedback for the authors:1. Figure 2D: Why do replisome numbers increase with increasing temperature and how does this observation reconcile with the increased speeds as well? Do both replication fork numbers and speed of individual replisomes contribute to the observed growth rates across temperatures?

We thank the Reviewer for this comment. The key point is that, at increasing temperature, the average replisome speed does not increase as fast as the growth rate, as it can be seen from Figure 2c in the manuscript. Therefore, multiple fork replication is necessary to fill this gap. In conclusion, both replication fork numbers and speed of individual replisomes provide a significant contribution, as the Reviewer suggests. We clarify this point in the revised draft.

2. Arguments in Appendix 2 and L140-150 are unclear and this reviewer is unable to understand the context of the same.

We rephrased the argument to make it clearer.

3. Did the authors consider sequencing a dnaQ proof reading mutant (PMID: 8610131) to test the speed vs accuracy tradeoff hypothesis?

We thank the Reviewer for suggesting this test. We agree that this would be an interesting experiment. However, we feel that this beyond the scope of our manuscript and more appropriate for a future study.

4. The Discussion section needs to be more focused on what the main conclusions of this work are. For example, the discussion on replication termination is not very informative. Instead, it would have been helpful if the authors could have explained how they envision the impact of temperature on replisome speed and cell cycle progression, a relationship that seems distinct from the impact of media/ nutrient on these dynamics. Secondly, do mutation rates also increase with increasing temperature, and do the authors envision mutation hotspots given their periodic positional data on the replisome? If so, what determines this periodicity? In the current manuscript, these ideas are not synthesized in a coherent fashion, that makes it harder to appreciate the unique findings.

We thank the Reviewer for this comment. We revised the Discussion section, that is now more focused on the conclusion of this work as requested. In particular, we expand on the impact of temperature of replisome speed and the implication for bacterial physiology. Other aspects of our results are also discussed more extensively, for example the hypotheses on the causes of the oscillations. We did not expand on the behavior of the mutation rate, as we felt that this falls beyond the direct implications of our work.

[Editors' note: further revisions were suggested prior to acceptance, as described below.]

The manuscript has been improved but there are some remaining issues that need to be addressed, as outlined below:As noted by reviewer #1, the statement that the growth rate is in the steady state in LB is not very solid. Please address the concerns regarding this issue. Unless the authors feel that it is necessary, we do not require additional experiments. However, the authors should1. Clarify throughout the manuscript the nuances regarding the lack of stationarity in LB.

We thank the Editor for highlighting this point in the reviewer’s comment. In the revised version, we clarify that in LB medium we cannot be sure that our cultures are at steady growth.

Importantly, although our model assumes steady growth for simplicity, this assumption is not crucial for our results. We prove this point by solving our model in the case of a fork firing rate that declines with time, thus violating the steady-state assumption.

The solution of this model shows that relaxing the steady-state approximation only leads to minor changes in the DNA abundance distribution, as shown in the new Appendix 7 in the revised manuscript.

2. Improve the growth figures to better show the dependence of growth rate on temperature, and whether the populations are in the exponential growth phase.

We agree that providing individual growth curves can help to appreciate whether the populations are in the exponential growth phase. Following this suggestion, we now include average growth curves for each temperature (without rescaling with the growth rate) in Figure 5a of the revised manuscript. We show the exponential fits corresponding to each temperature in the same figure. In addition, we provided growth curves for each different samples at each temperature in Figure 5—Figure supplement 1.

3. Further elaborate if the oscillations could be due to the fact that the population may have not reached the steady state in LB.

As mentioned in point 1., the new Appendix 7. presents a solution of a model that does not assume a steady state. We instead assume that the fork firing rate *k* declines with time as a consequence of nutrient depletion. We find that the DNA abundance predicted by this model presents only minor differences from that predicted by the model assuming steady growth.

In particular, we find that the discrepancy between these two models is not compatible with the shape as the oscillations of DNA abundance that we observe in the data. Most importantly, this discrepancy is much smaller than the magnitude of the observed oscillations. This result supports that the oscillations are not an artefact of the possible lack of a well-defined steady-state.

We do, however, agree that it will be important to verify our result in different kinds of media, and mention this point in the revised Discussion.

Reviewer 1 (Recommendations for the authors):I feel the requests and suggestions which were presented to the authors to revise the presented manuscript were very reasonable However, after reading the revised manuscript and the responses provided by the authors, I now feel that the authors did not even do the bare minimum to address the raised concerns. This particularly concerns a major concern about steady state growth which I raised, and which have been discussed away very carelessly by the authors. I do not support publication until this issue is honestly addressed and fixed in the manuscript as it relates to the major finding promoted by the authors: oscillating replication speeds at fast growth.In their reply the authors commented on my concern about the steady growth assumption in LB and state that their ‘medium leads to well-defined steady growth rates’. I think this statement is simply false. The fact that a growth-rate can be reproducibly obtained is not sufficient to prove that cells are in a steady state. For that one needs to show that the same growth-rate is obtained over several generations. Why is being picky on this issue so important? Because the authors assume a steady state in their method to infer replication speeds (with e.g. the firing rate being equal the growth rate etc). But it really remains unclear how well the inference method works in cultures which adjust to changing conditions and which are thus not in a steady state. This is a real concern for growth in LB which the authors chose as major growth condition. It is impossible to obtain a good steadily growing culture in LB as the availability of amino acids and other nutrient sources is drastically changing. This can for example be seen in growth curves where the density clearly derivates from an exponential fit when analyzing growth over several doublings. So with this the question remains: are the oscillations the authors report real or are they rather an artifact of falsely assuming steady state growth when doing their analysis. This concerns is highlighted even further by the newly presented findings stating that oscillations in speed are not observed for slower growth in minimal medium. So are these oscillations only observed at fast growth in LB because oscillations in speed are a biological reality at fast growth, or are they observed because a steady state assumption has been falsely assumed? I really want to be constructive here but feel the authors must address this concern very rigorously before this work should be published. So let me try to phrase my concerns in even more detail and in the context of the author's reply

We thank the Reviewer for his/her comments. We apologize for having misinterpreted his/her major concern, which has lead to a revision that did not fully address this concern.

Let us provide a general answer to the Reviewer’s comments and then enter the more specific points in the following. First of all, our individual growth curves are very well fitted by a logistic function. This function serves to describe both the exponential regime and the consequent saturation to the stationary phase, as customarily done and as discussed in (Zwietering et al. 1990). In the new revision, we provide additional figures showing fits of individual curves and summary curves at different temperatures, making it possible to appreciate the quality of the data. In all cases, we harvested cells during the exponential growth phase. We also remark that the cultures used to inoculate medium were previously grown in fresh medium until reaching an OD600 of about 1.0 at the target temperature.

In summary, our results show that our growth curves are very reproducible and very well characterized by a logistic growth, i.e. an exponential phase followed by a stationary phase. The exponential phase lasts for about 4 generations at all temperatures. On the other hand, although this exponential phase was preceded by another exponential growth, we agree with the Reviewer that we cannot be sure that cells are in a steady growth regime.

To address this issue, we generalized our model to a situation in which the fork firing rate is time-dependent and, in particular, slows down with time following nutrient depletion. These new results show that relaxing the assumption of steady growth has very limited impact on our results. In particular, the deviations we see from the steady model are far too small to explain the oscillations we see in our data.

In conclusion, we appreciate the issue raised by the Reviewer and agree that future experiments with simpler media will be important. However, our new theoretical results show that assuming near-exponential growth for a few generations is sufficient for our aims, and support the idea that the oscillations we observe are a real phenomenon.

To address my concern the authors simply showed the growth-curves in Figure 5 and stated that this supports cultures are in exponential steady growth. However, the presented analysis is not sufficient to prove steady state. For example, there are not exponential fits shown which prove that growth-rates are maintained for reasonable ranges of density and time. In fact, when looking at the presented panel 5a it appears that the growth-curves at different temperatures are pretty much comparable, while a more detailed fit presumably used to infer growth-rate values (instantaneous rates at a specific OD/timepoint) suggest that growth-rates are different and this difference is what the authors use in the second panel of that figure and to discuss the relation between growth rate and replication speed. Given what is known about growth on LB a zoomed in version of these growth curves will reveal that growth is not really exponential for a reasonable long amount of time (say dozens of generations until protein levels reach a steady state) and as such no steady state have been obtained. In this context I strongly believe that a more careful experimentation with a well-defined steady states in a well-defined media (e.g. rich defined media with exponential growth and the some growth rates observed over > 10 doublings) would have been way better to establish that the inferred oscillations in speed are a real biological phenomenon. I understand that the authors are reluctant to add such additional experiments and I am not insisting on them. However, the concern needs to be taken seriously. I think the authors should:1) Clearly state in the manuscript that the presented measurements have not been taken during a well-defined steady state

In the revised version, we clearly state at the end of the Results section that, although the growth curves appear as exponential, we cannot be sure that the cultures have reached a regime of steady growth. Then, to address this shortcoming, we relax the assumption of steady growth in the model. Our results, presented in the new Appendix 7, show that assuming a near-constant fork firing rate for about 4 generations (as we observe) is sufficient to obtain DNA abundance curves that are very similar to those of the steady-state model. This result indicates that the possible lack of steady growth in LB medium should not significantly affect our conclusions.

2) Plot and analyze better the growth-curves such that the growth behavior and the differences in growth for different temperatures become visible). In such a plot exponential fits to the growth curves should also be shown which directly allow the reader to compare how well-defined and distinct the growth-rates really are.

We agree with the Reviewer that providing individual growth curves can help to appreciate whether the populations are in the exponential growth phase. Following this suggestion, we now include average growth curves for each temperature (without rescaling with the growth rate) in Figure 5a of the revised manuscript. We also show the exponential fits corresponding to each temperature in the same figure. In addition, we provided growth curves for each different sample at each temperature in Figure 5—figure supplement 1. We thank the Reviewer for suggesting this addition that should provide the reader with all the necessary information.

3) Raise in the conclusion of the paper that there is a possibility that oscillations might occur because the cultures grew in LB and have thus not reached steady state.

Our theoretical results do not support the idea that oscillations might be caused by the fact that in LB medium cultures have not reached steady state. In the revised conclusion, we do however mention this issue and suggest that future studies in different media will be important.

4) Include some sort of error propagation analysis which shows how strong the errors in replication speed can be when assuming a culture not yet in steady growth but running the analysis to infer steady state with the assumption of a steady state. For example, assuming the number of cells with different replication forks is still slowly adjusting with a certain timescale and assuming the replication speed is constant, use a fully dynamical model to derive a DNA abundance distribution. Then inferring the speed by analysis the distribution (using the presented approach assuming steady state).

As stated in the answer to point (1), we studied a variant of the model which does not assume steady state to address this point. This analysis is presented in the new Appendix 7.

We find that this model leads to an inference of the replisome speed which is very close to that of the original steady-state model. Further, the discrepancy between these two models is not consistent with the shape of the oscillations of DNA abundance that we observe. Most importantly, this discrepancy is much smaller than the magnitude of the oscillations of our data. This result indicates that the oscillations are not caused by a possible lack of a well-defined steady-state.